# DRAGONDIFFUSION: ENABLING DRAG-STYLE MANIPULATION ON DIFFUSION MODELS

**Chong Mou**[1,3]    **Xintao Wang**[2]    **Jiechong Song**[1]    **Ying Shan**[2]    **Jian Zhang**[1,3*]

[1]School of Electronic and Computer Engineering, Shenzhen Graduate School, Peking University
[2]ARC Lab, Tencent PCG
[3]Peking University Shenzhen Graduate School-Rabbitpre AIGC Joint Research Laboratory
{eechongm, xintao.alpha}@gmail.com, {songjiechong, zhangjian.sz}@pku.edu.cn

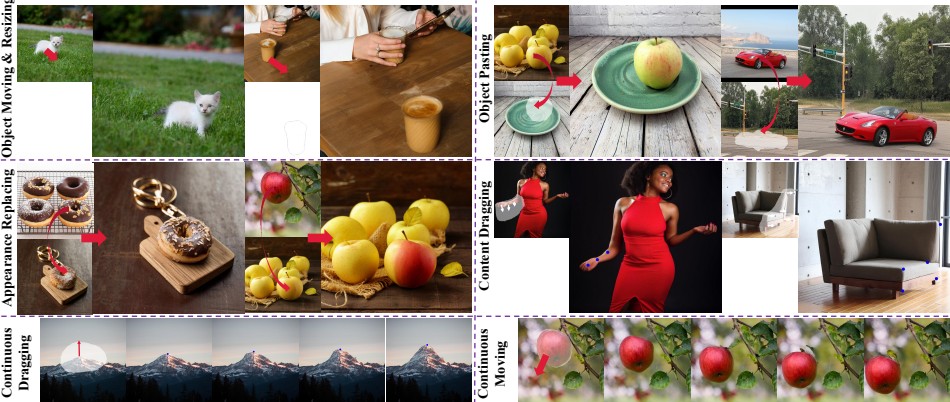

Figure 1: The image editing tasks that our DragonDiffusion can achieve without training.

## ABSTRACT

Despite the ability of text-to-image (T2I) diffusion models to generate high-quality images, transferring this ability to accurate image editing remains a challenge. In this paper, we propose a novel image editing method, **DragonDiffusion**, enabling **Drag**-style manipulation **on Diffusion** models. Specifically, we treat image editing as the change of feature correspondence in a pre-trained diffusion model. By leveraging feature correspondence, we develop energy functions that align with the editing target, transforming image editing operations into gradient guidance. Based on this guidance approach, we also construct multi-scale guidance that considers both semantic and geometric alignment. Furthermore, we incorporate a visual cross-attention strategy based on a memory bank design to ensure consistency between the edited result and original image. Benefiting from these efficient designs, all content editing and consistency operations come from the feature correspondence without extra model fine-tuning. Extensive experiments demonstrate that our method has promising performance on various image editing tasks, including within a single image (*e.g.*, object moving, resizing, and content dragging) or across images (*e.g.*, appearance replacing and object pasting). Code is available at https://github.com/MC-E/DragonDiffusion.

# 1 INTRODUCTION

Thanks to the large-scale training data and huge computing power, generative models have developed rapidly, especially text-to-image (T2I) diffusion models Saharia et al. (2022); Rombach et al. (2022); Nichol et al. (2022); Ramesh et al. (2022), which aims to generate images conditioned on a given text/prompt. However, this generative capability is usually diverse, and it is challenging to design suitable prompts to generate images consistent with what the user has in mind Mou et al. (2023); Zhang et al. (2023), let alone fine-grained image editing based on the text condition.

---

*Corresponding author

This work was supported by National Natural Science Foundation of China under Grant 62372016.

In the community of image editing, previous methods are usually designed based on GANs Abdal et al. (2019; 2020); Alaluf et al. (2022) due to the compact and editable latent space, *e.g.*, the $\mathcal{W}$ space in StyleGAN Karras et al. (2019). Recently, DragGAN Pan et al. (2023) proposes a point-to-point dragging scheme, which can achieve refined content dragging. However, it is limited by the capacity and generalization of GANs. Compared to GANs, diffusion model Ho et al. (2020) has higher stability and superior generation quality. Due to the lack of a concise and editable latent space, numerous diffusion-based image editing methods Hertz et al. (2022); Feng et al. (2022); Balaji et al. (2022) are built based on T2I diffusion models via correspondence between text and image features. Recently, self-guidance Epstein et al. (2024) proposes a differentiable approach that employs cross-attention maps between text and image to locate and calculate the size of objects within images. Then, gradient guidance is utilized to edit these properties. However, the correspondence between text and image features is weak, heavily relying on the design of prompts. Moreover, in complex or multi-object scenarios, text struggles to build accurate correspondence with a specific object. In this paper, we aim to investigate whether the diffusion model can achieve drag-style image editing, which is a fine-grained and generalized editing ability not limited to point dragging.

In the large-scale T2I diffusion model, besides the correspondence between text features and intermediate image features, there is also a strong correspondence across image features. This characteristic is studied in DIFT Tang et al. (2023), which demonstrates that this correspondence is high-level, enabling point-to-point correspondence of relevant image content. Therefore, we are intrigued by the possibility of utilizing this strong correspondence across image features to achieve image editing. In this paper, we regard image editing as the change of feature correspondence and convert it into gradient guidance via energy functions Dhariwal & Nichol (2021) in score-based diffusion Song et al. (2020b). Additionally, the content consistency between editing results and original images is also ensured by feature correspondence in a visual cross-attention design. Here, we notice that there is a concurrent work, DragDiffusion Shi et al. (2023), studying this issue. It uses LORA Ryu (2023) to maintain consistency with the original image and optimizes the latent in a specific diffusion step to perform point dragging. Unlike DragDiffusion, our image editing is achieved by energy functions and a visual cross-attention design, without extra model fine-tuning or new blocks. In addition, we can complete various drag-style image editing tasks beyond the point dragging, as shown in Fig. 1.

In summary, the contributions of this paper are as follows:

- We achieve drag-style image editing via image feature correspondence in the pre-trained diffusion model. We also study the roles of the features in different layers and develop multi-scale guidance that considers both semantic and geometric correspondence.

- We design a memory bank, further utilizing the image feature correspondence to maintain the consistency between editing results and original images. In conjunction with gradient guidance, our method allows a direct transfer of T2I generation ability in diffusion models to image editing tasks without the need for extra model fine-tuning or new blocks.

- Extensive experiments demonstrate that our method has promising performance in various image editing tasks, including editing within a single image (*e.g.*, object moving, resizing, and content dragging) or across images (*e.g.*, appearance replacing and object pasting).

## 2 RELATED WORK

### 2.1 DIFFUSION MODELS

Recently, diffusion models Ho et al. (2020) have achieved great success in the community of image synthesis. It is designed based on thermodynamics Sohl-Dickstein et al. (2015); Song & Ermon (2019), including a diffusion process and a reverse process. In the diffusion process, a natural image $\mathbf{x}_0$ is converted to a Gaussian distribution $\mathbf{x}_T$ by adding random Gaussian noise with $T$ iterations. The reverse process is to recover $\mathbf{x}_0$ from $\mathbf{x}_T$ by several denoising steps. Therefore, the diffusion model is to train a denoiser, conditioned on the current noisy image $\mathbf{x}_t$ and time step $t$:

$$\mathbb{E}_{\mathbf{x}_0, t, \boldsymbol{\epsilon}_t \sim \mathcal{N}(0,1)} \left[ ||\boldsymbol{\epsilon}_t - \epsilon_{\boldsymbol{\theta}}(\mathbf{x}_t, t)||_2^2 \right], \tag{1}$$

where $\epsilon_{\boldsymbol{\theta}}$ is the function of the denoiser. Recently, some text-conditioned diffusion models (*e.g.*, GLID Nichol et al. (2022) and StableDiffusion(SD) Rombach et al. (2022)) are proposed. Especially SD, transforming $\mathbf{x}_t$ to the latent space $\mathbf{z}_t$, significantly improves the generation performance. From

the continuous perspective Song et al. (2020b), diffusion models can be viewed as a score function (*i.e.*, $\epsilon_{\boldsymbol{\theta}}(\mathbf{x}_t, t) \approx \nabla_{\mathbf{x}_t} \log q(\mathbf{x}_t)$) that samples from the corresponding distribution Song & Ermon (2020) according to Langevin dynamics Sohl-Dickstein et al. (2015); Song & Ermon (2019).

## 2.2 ENERGY FUNCTION IN DIFFUSION MODEL

From the continuous perspective of score-based diffusion, the external condition $\mathbf{y}$ can be combined by a conditional score function, *i.e.*, $\nabla_{\mathbf{x}_t} \log q(\mathbf{x}_t|\mathbf{y})$, to sample from a more enriched distribution. The conditional score function can be further decomposed as:

$$\nabla_{\mathbf{x}_t} \log q(\mathbf{x}_t|\mathbf{y}) = \nabla_{\mathbf{x}_t} \log \left( \frac{q(\mathbf{y}|\mathbf{x}_t)q(\mathbf{x}_t)}{q(\mathbf{y})} \right) \propto \nabla_{\mathbf{x}_t} \log q(\mathbf{x}_t) + \nabla_{\mathbf{x}_t} \log q(\mathbf{y}|\mathbf{x}_t), \quad (2)$$

where the first term is the unconditional denoiser, and the second term refers to the conditional gradient produced by an energy function $\mathcal{E}(\mathbf{x}_t; t, \mathbf{y}) = q(\mathbf{x}_t|\mathbf{y})$. $\mathcal{E}$ can be selected based on the generation target, such as a classifier Dhariwal & Nichol (2021) to specify the category of generation results. Energy function has been used in various controllable generation tasks, *e.g.*, sketch-guided generation Voynov et al. (2023), mask-guided generation Singh et al. (2023), universal guidance Yu et al. (2023); Bansal et al. (2023), and image editing Epstein et al. (2024). These methods, inspire us to transform editing operations into conditional gradients, achieving fine-grained image editing.

## 2.3 IMAGE EDITING

In image editing, numerous previous methods Abdal et al. (2019; 2020); Alaluf et al. (2022) invert images into the latent space of StyleGAN Karras et al. (2019) and then edit the image by manipulating latent vectors. Motivated by the success of diffusion model Ho et al. (2020), various diffusion-based image editing methods Avrahami et al. (2022); Hertz et al. (2022); Kawar et al. (2023); Meng et al. (2021); Brooks et al. (2023) are proposed. Most of them use text as the editing control. For example, Kawar et al. (2023); Valevski et al. (2023); Kwon & Ye (2022) perform model fine-tuning on a single image and then generate the editing result by target text. Prompt2Prompt Hertz et al. (2022) achieves specific object editing by exchanging text-image attention maps. SDEdit Meng et al. (2021) performs image editing by adding noise to the original image and then denoising under new text conditions. InstructPix2Pix Brooks et al. (2023) finetunes the diffusion model with text as the editing instruction. Recently, Self-guidance Epstein et al. (2024) transforms image editing operations into gradients through the correspondence between text and image features. However, the correspondence between text and image is weak, unable to perform fine-grained editing. Recently, DragGAN Pan et al. (2023) presents a point-to-point dragging scheme. Nevertheless, its editing quality and generalization are limited by GANs. How to utilize the high-quality and diverse generation ability of diffusion models for fine-grained image editing is still an open challenge.

## 3 METHOD

### 3.1 PRELIMINARY: HOW TO CONSTRUCT ENERGY FUNCTION IN DIFFUSION

Modeling an energy function $\mathcal{E}(\mathbf{x}_t; t, \mathbf{y})$ to produce the conditional gradient $\nabla_{\mathbf{x}_t} \log q(\mathbf{y}|\mathbf{x}_t)$ in Eq. 2, remains an open question. $\mathcal{E}$ measures the distance between $\mathbf{x}_t$ and the condition $\mathbf{y}$. Some methods Dhariwal & Nichol (2021); Voynov et al. (2023); Zhao et al. (2022) train a time-dependent distance measuring function, *e.g.*, a classifier Dhariwal & Nichol (2021) to predict the probability that $\mathbf{x}_t$ belongs to category $\mathbf{y}$. However, the training cost and annotation difficulty are intractable in our image editing task. Some tuning-free methods Yu et al. (2023); Bansal et al. (2023) propose using the clean image $\mathbf{x}_{0|t}$ predicted at each time step $t$ to replace $\mathbf{x}_t$ for distance measuring, *i.e.*, $\mathcal{E}(\mathbf{x}_t; t, \mathbf{y}) \approx \mathcal{D}(\mathbf{x}_{0|t}; t, \mathbf{y})$. Nevertheless, there is a bias between $\mathbf{x}_{0|t}$ and $\mathbf{x}_0$, and there is hardly a suitable $\mathcal{D}$ for distance measuring in image editing tasks. Hence, the primary issue is whether we can circumvent the training requirement and construct an energy function to measure the distance between $\mathbf{x}_t$ and the editing target. Recent work Tang et al. (2023) has shown that the feature correspondence in the diffusion UNet-denoiser $\epsilon_\theta$ is high-level, enabling point-to-point correspondence measuring. Inspired by this characteristic, we propose reusing $\epsilon_\theta$ as a tuning-free energy function to transform image editing operations into the change of feature correspondence.

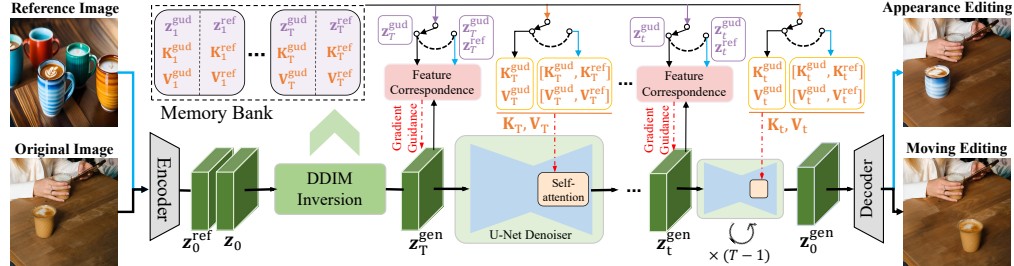

Figure 2: Overview of our DragonDiffusion, containing a memory bank and score-based gradient guidance on the pre-trained SD Rombach et al. (2022) without extra training or modules.

## 3.2 OVERVIEW

The editing objective of our DragonDiffusion involves two issues: changing the content to be edited and preserving unedited content. For example, if a user wants to move the cup in an image, the generated result only needs to change the position of the cup, while the appearance of the cup and other unedited content should not change. An overview of our method is presented in Fig. 2, which is built on the pre-trained SD Rombach et al. (2022) to support image editing with and without reference images. Since SD is a latent diffusion model (LDM), we first encode the original image $\mathbf{x}_0$ into the latent space $\mathbf{z}_0$, which is then reversed to $\mathbf{z}_T$ by DDIM inversion Song et al. (2020a). If the reference image $\mathbf{x}_0^{ref}$ exists, it will also be involved in the inversion to produce $\mathbf{z}_T^{ref}$. In this process, we store some intermediate features and latent at each time step to build a memory bank, which is used to provide guidance for subsequent image editing. In generation, we transform the information stored in the memory bank into content editing and consistency guidance through two paths, *i.e.*, visual cross-attention and gradient guidance. Both of these paths are built based on feature correspondence in the pre-trained SD, without extra model fine-tuning or new blocks.

## 3.3 DDIM INVERSION WITH MEMORY BANK

In our image editing process, the starting point $\mathbf{z}_T$, produced by DDIM inversion Song et al. (2020a), can provide a good generation prior to maintain consistency with the original image. However, relying solely on the final step of this approximate inversion can hardly provide accurate generation guidance. Therefore, we fully utilize the information in DDIM inversion by building a memory bank to store the latent $\mathbf{z}_t^{gud}$ at each inversion step $t$, as well as corresponding keys $\mathbf{K}_t^{gud}$ and values $\mathbf{V}_t^{gud}$ in the self-attention module of the decoder within the UNet denoiser. Note that in some cross-image editing tasks (*e.g.*, appearance replacing, object pasting), reference images are required. In these tasks, the memory bank needs to be doubled to store the information of the reference images. Here, we utilize $\mathbf{z}_t^{ref}$, $\mathbf{K}_t^{ref}$, and $\mathbf{V}_t^{ref}$ to represent them. The information stored in the memory bank will provide more accurate guidance for the subsequent image editing process.

## 3.4 GRADIENT-GUIDANCE-BASED EDITING DESIGN

Inspired by classifier guidance Dhariwal & Nichol (2021), we build energy functions to transform image editing operations into gradient guidance in diffusion sampling. An intuitive illustration is presented in Fig. 3, showing a continuous sampling space of the score-based diffusion Song et al. (2020b). The sampling starting point $\mathbf{z}_T$, obtained from DDIM inversion, will approximately return to the original point only according to the gradient/score predicted by the denoiser. After incorporating the gradient guidance generated by the energy function that matches the editing target, the additional guidance gradient will change the path to reach a sampling result that meets the editing target.

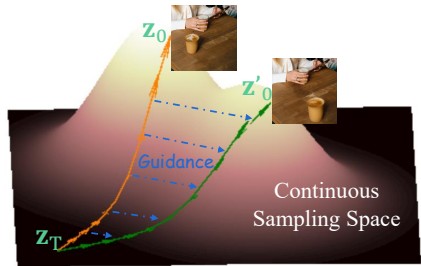

Figure 3: Illustration of continuous sampling space in score-based diffusion. Bright colors indicate areas where target data is densely distributed. The orange and green paths respectively refer to the diffusion paths without and with external gradient guidance.

### 3.4.1 ENERGY FUNCTION VIA FEATURE CORRESPONDENCE

In our DragonDiffusion, energy functions are designed to provide gradient guidance for image editing, mainly including content editing and consistency terms. Specifically, at the $t$-th time step, we reuse the UNet denoiser $\epsilon_{\boldsymbol{\theta}}$ to extract intermediate features $\mathbf{F}_t^{gen}$ from the latent $\mathbf{z}_t^{gen}$ at the current time step. The same operation is used to extract guided features $\mathbf{F}_t^{gud}$ from $\mathbf{z}_t^{gud}$ in memory bank. Following DIFT Tang et al. (2023), $\mathbf{F}_t^{gen}$ and $\mathbf{F}_t^{gud}$ come from intermediate features in the UNet decoder. The image editing operation is represented by two binary masks (*i.e.*, $\mathbf{m}^{gud}$ and $\mathbf{m}^{gen}$) to locate the original content position and target dragging position, respectively. Therefore, the energy function is built by constraining the correspondence between these two regions in $\mathbf{F}_t^{gud}$ and $\mathbf{F}_t^{gen}$. Here, we utilize cosine distance $\cos(\cdot) \in [-1, 1]$ to measure the similarity and normalize it to $[0, 1]$:

$$\mathcal{S}_{local}(\mathbf{F}_t^{gen}, \mathbf{m}^{gen}, \mathbf{F}_t^{gud}, \mathbf{m}^{gud}) = 0.5 \cdot \cos\left(\mathbf{F}_t^{gen}[\mathbf{m}^{gen}], \ \text{sg}(\mathbf{F}_t^{gud}[\mathbf{m}^{gud}])\right) + 0.5, \quad (3)$$

where $\text{sg}(\cdot)$ is the gradient clipping operation. Eq. 3 is mainly used for dense constraints on the spatial location of content. In addition, a global appearance similarity is defined as:

$$\mathcal{S}_{global}(\mathbf{F}_t^{gen}, \mathbf{m}^{gen}, \mathbf{F}_t^{gud}, \mathbf{m}^{gud}) = 0.5 \cdot \cos\left(\frac{\sum \mathbf{F}_t^{gen}[\mathbf{m}^{gen}]}{\sum \mathbf{m}^{gen}}, \ \text{sg}(\frac{\sum \mathbf{F}_t^{gud}[\mathbf{m}^{gud}]}{\sum \mathbf{m}^{gud}})\right) + 0.5, \quad (4)$$

which utilizes the mean of the features in a region as a global appearance representation. When we want to have fine control over the spatial position of an object or a rough global control over its appearance, we only need to constrain the similarity in Eq. 3 and Eq. 4 to be as large as possible. Therefore, the energy function to produce editing guidance is defined as:

$$\mathcal{E}_{edit} = \frac{1}{\alpha + \beta \cdot \mathcal{S}(\mathbf{F}_t^{gen}, \mathbf{m}^{gen}, \mathbf{F}_t^{gud}, \mathbf{m}^{gud})}, \quad \mathcal{S} \in \{\mathcal{S}_{local}, \ \mathcal{S}_{global}\}, \quad (5)$$

where $\alpha$ and $\beta$ are two hyper-parameters, which are set as 1 and 4, respectively. In addition to editing, we hope the unedited content remains consistent with the original image. We use a mask $\mathbf{m}^{share}$ to locate areas without editing. The similarity between the editing result and the original image in $\mathbf{m}^{share}$ can also be calculated by the cosine similarity as $\mathcal{S}_{local}(\mathbf{F}_t^{gen}, \mathbf{m}^{share}, \mathbf{F}_t^{gud}, \mathbf{m}^{share})$. Therefore, the energy function to produce content consistency guidance is defined as:

$$\mathcal{E}_{content} = \frac{1}{\alpha + \beta \cdot \mathcal{S}_{local}(\mathbf{F}_t^{gen}, \mathbf{m}^{share}, \mathbf{F}_t^{gud}, \mathbf{m}^{share})}. \quad (6)$$

In addition to $\mathcal{E}_{edit}$ and $\mathcal{E}_{content}$, an optional guidance term $\mathcal{E}_{opt}$ may need to be added in some tasks to achieve the editing goal. Finally, the base energy function is defined as:

$$\mathcal{E} = w_e \cdot \mathcal{E}_{edit} + w_c \cdot \mathcal{E}_{content} + w_o \cdot \mathcal{E}_{opt}, \quad (7)$$

where $w_e$, $w_c$, and $w_o$ are hyper-parameters to balance these guidance terms. They vary slightly in different editing tasks but are fixed within the same task. Finally, regarding $[\mathbf{m}^{gen}, \mathbf{m}^{share}]$ as condition, the conditional score function in Eq. 2 can be written as:

$$\nabla_{\mathbf{z}_t^{gen}} \log q(\mathbf{z}_t^{gen}|\mathbf{y}) \propto \nabla_{\mathbf{z}_t^{gen}} \log q(\mathbf{z}_t^{gen}) + \nabla_{\mathbf{z}_t^{gen}} \log q(\mathbf{y}|\mathbf{z}_t^{gen}), \quad \mathbf{y} = [\mathbf{m}^{gen}, \mathbf{m}^{share}]. \quad (8)$$

The conditional gradient $\nabla_{\mathbf{z}_t^{gen}} \log q(\mathbf{y}|\mathbf{z}_t^{gen})$ can be computed by $\nabla_{\mathbf{z}_t^{gen}} \mathcal{E}$, which will also multiplies by a learning rate $\eta$. In experiments, we find that the gradient guidance in later diffusion generation steps hinders the generation of textures. Therefore, we only add gradient guidance in the first $n$ steps of diffusion generation. Experientially, we set $n = 30$ in 50 sampling steps.

### 3.4.2 MULTI-SCALE FEATURE CORRESPONDANCE

The decoder of the UNet denoiser contains four blocks of different scales. DIFT Tang et al. (2023) finds that the second layer contains more semantic information, while the third layer contains more geometric information. We also studied the role of features from different layers in image editing tasks, as shown in Fig. 4. In the experiment, we set $\mathbf{z}_T$ as random Gaussian noise and set $\mathbf{m}^{gen}$, $\mathbf{m}^{gud}$ as zeros matrixes. $\mathbf{m}^{share}$ is set as a ones matrix. In this way, generation relies solely on content consistency guidance (*i.e.*, Eq. 6) to restore image content. We can find that the guidance

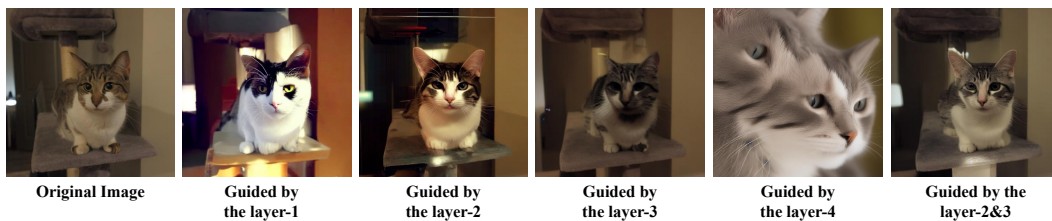

| Original Image | Guided by the layer-1 | Guided by the layer-2 | Guided by the layer-3 | Guided by the layer-4 | Guided by the layer-2&3 |

Figure 4: Illustration of using features from different layers as guidance to restore the original image. $\mathbf{z}_T$ is randomly initialized. The generation is solely guided by content consistency guidance in Eq. 6.

from the first layer is too high-level to reconstruct the original image accurately. The guidance from the fourth layer has weak feature correspondence, resulting in significant differences between the reconstructed and original images. The features from the second and third layers are more suitable to produce guidance signals, and each has its own specialty. Concretely, the features in the second layer contain more semantic information and can reconstruct images that are semantically similar to the original image but with some differences in content details. The features in the third layer tend to express low-level characteristics, but they cannot provide effective supervision for high-level texture, resulting in blurry results. In our design, we combine these two levels (*i.e.*, high and low) of guidance by proposing a multi-scale supervision approach. Specifically, we compute gradient guidance on the second and third layers. The reconstructed results in Fig. 4 also demonstrate that this combination can balance the generation of low-level and high-level visual characteristics.

### 3.4.3 IMPLEMENTATION DETAILS FOR EACH APPLICATION

**Object moving.** In the task of object moving, $\mathbf{m}^{gen}$ and $\mathbf{m}^{gud}$ locate the same object in different spatial positions. $\mathbf{m}^{share}$ is the complement ($\mathbf{C}_u$) of the union ($\cup$) of $\mathbf{m}^{gen}$ and $\mathbf{m}^{gud}$, *i.e.*, $\mathbf{m}^{share} = \mathbf{C}_u(\mathbf{m}^{gen} \cup \mathbf{m}^{gud})$. However, solely using the content editing and consistency guidance in Eq. 5 and Eq. 6 can lead to some issues, as shown in the second image of Fig. 5. Concretely, although the bread is moved according to the editing signal, some of the bread content is still preserved in its original position in the generated

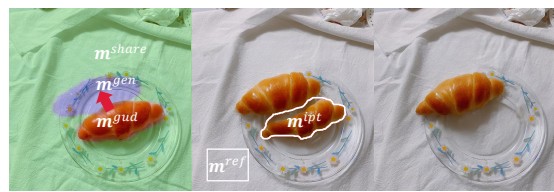

| Original Image | Moving w/o $\varepsilon_{opt}$ | Moving w $\varepsilon_{opt}$ |

Figure 5: Visualization of the effectiveness of inpainting guidance ($\mathcal{E}_{opt}$) in the object moving task, presenting that $\mathcal{E}_{opt}$ can guide the inpainting of the area where the object is initially located.

result. This is because the energy function does not constrain the area where the moved object was initially located, causing inpainting to easily restore the original object. To rectify this issue, we use the optional energy term (*i.e.*, $\mathcal{E}_{opt}$ in Eq. 7) to constrain the inpainting content to be dissimilar to the moved object and similar to a predefined reference region. Here, we use $\mathbf{m}^{ref}$ to locate the reference region and define $\mathbf{m}^{ipt} = \{p | p \in \mathbf{m}^{gud} \text{ and } p \notin \mathbf{m}^{gen}\}$ to locate the inpainting region. Finally, $\mathcal{E}_{opt}$ in this task is defined as:

$$\mathcal{E}_{opt} = \frac{w_i}{\alpha + \beta \cdot \mathcal{S}_{global}(\mathbf{F}_t^{gen}, \mathbf{m}^{ipt}, \mathbf{F}_t^{gud}, \mathbf{m}^{ref})} + \mathcal{S}_{local}(\mathbf{F}_t^{gen}, \mathbf{m}^{ipt}, \mathbf{F}_t^{gud}, \mathbf{m}^{ipt}), \quad (9)$$

where $w_i$ is a weight parameter, set as $2.5$ in our implementation. The third image in Fig. 5 shows that this design can effectively achieve the editing goal without impeachable artifact.

**Object resizing.** The score function in this task is the same as the object moving, except that a scale factor $\gamma > 0$ is added during feature extraction. Specifically, we use interpolation to transform $\mathbf{m}^{gud}$ and $\mathbf{F}_t^{gud}$ to the target size, and then extract $\mathbf{F}_t^{gud}[\mathbf{m}^{gud}]$ as the feature of the resized object. To locate the target object with the same size in $\mathbf{F}_t^{gen}$, we resize $\mathbf{m}^{gen}$ with the same scale factor $\gamma$. Then we extract a new $\mathbf{m}^{gen}$ of the original size from the center of the resized $\mathbf{m}^{gen}$. Note that if $\gamma < 1$, we use 0 to pad the vacant area.

**Appearance replacing.** This task aims to replace the appearance between objects of the same category across images. Therefore, the capacity of the memory bank needs to be doubled to store extra information from the image containing the reference appearance, *i.e.*, $\mathbf{z}_t^{ref}$, $\mathbf{K}_t^{ref}$, and $\mathbf{V}_t^{ref}$. $\mathbf{m}^{gen}$ and $\mathbf{m}^{gud}$ respectively locate the editing object in the original image and the reference object

in the reference image. $\mathbf{m}^{share}$ is set as the complement of $\mathbf{m}^{gen}$, *i.e.*, $\mathrm{C_u}(\mathbf{m}^{gen})$. To constrain appearance, we choose $\mathcal{S}_{global}(\mathbf{F}_t^{gen}, \mathbf{m}^{gen}, \mathbf{F}_t^{ref}, \mathbf{m}^{gud})$ in Eq. 5. This task has no need for $\mathcal{E}_{opt}$.

**Object pasting.** Object pasting aims to paste an object from an image onto any position in another image. Although it can be completed by simple *copy-paste*, it often results in inconsistencies between the paste area and other areas due to differences in light and perspective, as shown in Fig. 6. As can be seen, the result obtained by *copy-paste* exists discontinuities, while the result generated by our DragonDiffusion can achieve a more harmonized integration of the scene and the pasted object. In implementation, similar to the appearance replacing, the memory bank needs to store information of the reference image, which contains the target object. $\mathbf{m}^{gen}$ and $\mathbf{m}^{gud}$ respectively mark the position of the object in the edited image and reference image. $\mathbf{m}^{share}$ is set as $\mathrm{C_u}(\mathbf{m}^{gen})$.

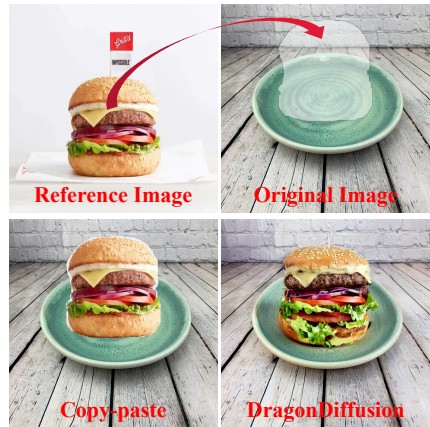

**Point dragging.** In this task, we want to drag image content via several points, as DragGAN Pan et al. (2023). In this case, $\mathbf{m}^{gen}$ and $\mathbf{m}^{gud}$ locate neighboring areas centered around the destination and starting points. Here, we

Figure 6: Visual comparison between our DragonDiffusion and direct *copy-paste* in cross-image object pasting.

extract a $3 \times 3$ rectangular patch centered around each point as the neighboring area. Unlike the previous tasks, $\mathbf{m}^{share}$ is manually defined.

## 3.5 Visual Cross-attention

As mentioned previously, two strategies are used to ensure the consistency between the editing result and the original image: (1) DDIM inversion to initialize $\mathbf{z}_T$; (2) content consistency guidance in Eq. 6. However, it is still challenging to maintain high consistency. Inspired by the consistency preserving in some video and image editing works Wu et al. (2022); Cao et al. (2023); Wang et al. (2023), we design a visual cross-attention guidance. Instead of generating guidance information through an independent inference branch, we reuse the intermediate features of the inversion process stored in the memory bank. Specifically, similar to the injection of text conditions in SD Rombach et al. (2022), we replace the key and value in the self-attention module of the UNet decoder with the corresponding key and value collected by the memory bank in DDIM inversion. Note that in the appearance replacing and object pasting tasks, the memory bank stores two sets of keys and values from the original image ($\mathbf{K}_t^{gud}, \mathbf{V}_t^{gud}$) and the reference image ($\mathbf{K}_t^{ref}, \mathbf{V}_t^{ref}$). In this case, we concatenate the two sets of keys and values in the length dimension. The visual cross-attention at each time step is defined as follows. ⓒ refers to the concatenation operation.

$$\begin{cases} \mathbf{Q}_t = \mathbf{Q}_t^{gen}; \ \mathbf{K}_t = \mathbf{K}_t^{gud} \text{ or } (\mathbf{K}_t^{gud}ⓒ\mathbf{K}_t^{ref}); \ \mathbf{V}_t = \mathbf{V}_t^{gud} \text{ or } (\mathbf{V}_t^{gud}ⓒ\mathbf{V}_t^{ref}) \\ \mathrm{Att}(\mathbf{Q}_t, \mathbf{K}_t, \mathbf{V}_t) = \mathrm{softmax}(\frac{\mathbf{Q}_t\mathbf{K}_t^T}{\sqrt{d}})\mathbf{V}_t. \end{cases} \quad (10)$$

## 4 Experiments

In experiments, we use StableDiffusion-V1.5 Rombach et al. (2022) as the base model. The inference adopts DDIM sampling with 50 steps, and we set the classifier-free guidance scale as 5.

## 4.1 Comparisons

In this part, we compare our DragonDiffusion with other methods on various image editing tasks.

**Content dragging**. In this task, we compare our method with the recent UserControllableLT Endo (2022), DragGAN Pan et al. (2023), and DragDiffusion Shi et al. (2023). We first present the time complexity of different methods in Tab. 1. Specifically, We divide the time complexity of different methods into two parts, *i.e.*, the preparing and inference stages. The preparing stage involves Diffusion/GAN inversion and model fine-tuning. The inference stage generates the editing result. The time complexity is tested on one point dragging, with the image resolution being $512 \times 512$.

Table 1: Quantitative evaluation on face manipulation with 68 and 17 points. The accuracy is calculated by Euclidean distance between edited points and target points. The initial distance (*i.e.*, *57.19* and *36.36*) is the upper bound, without editing. FID Seitzer (2020) is utilized to quantize the editing quality of different methods. The time complexity is computed on the '1 point' dragging.

| | Preparing complexity↓ | Inference complexity↓ | Unaligned face | 17 Points↓ From 57.19 | 68 Points↓ From 36.36 | FID↓ 17/68 points |
|---|---|---|---|---|---|---|
| UserControllableLT | **1.2**s | **0.05**s | ✗ | 32.32 | 24.15 | 51.20/50.32 |
| DragGAN | 52.40s | 6.71s | ✗ | **15.96** | **10.60** | 39.27/39.50 |
| DragDiffusion | 48.25s | 19.71s | ✓ | 22.95 | 17.32 | 38.06/36.55 |
| DragonDiffusion(ours) | 3.62s | 15.93s | ✓ | 18.51 | 13.94 | **35.75/34.58** |

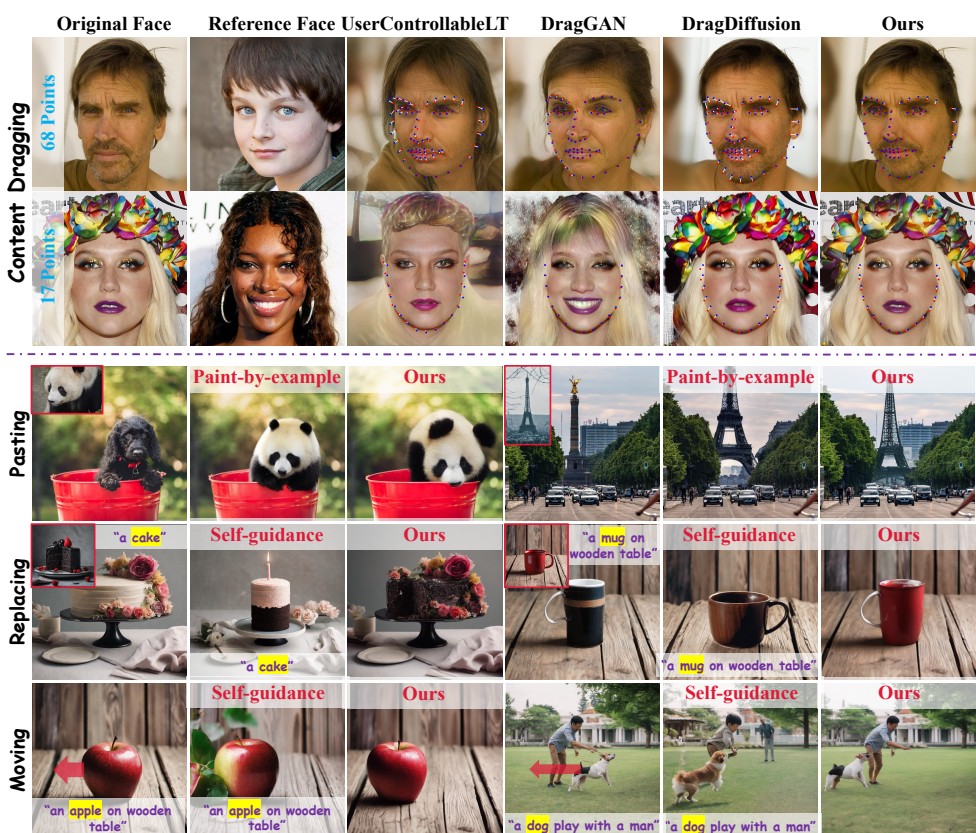

Figure 7: Qualitative comparison between our DragonDiffusion and other methods in face manipulation (target points are blue), object pasting, appearance replacing, and object moving.

The experiment is conducted on an NVIDIA A100 GPU with Float32 precision. The results present that our method is relatively efficient in the preparing stage, requiring only 3.62s to prepare $z_T$ and memory bank. The inference complexity is also acceptable for diffusion generation.

Following DragGAN Pan et al. (2023), the performance evaluation is conducted on the face keypoint manipulation with 17 and 68 points. The test set is randomly formed by 800 aligned faces from CelebA-HQ Karras et al. (2018) training set. Note that we do not set fixed regions for all methods, due to the difficulty in manually providing a mask for each face. In addition to accuracy, we also compute the FID Seitzer (2020) between face editing results and CelebA-HQ training set to represent the editing quality. The quantitative and qualitative comparison is presented in Tab. 1 and Fig. 7, respectively. One can see that our DragonDiffusion achieves promising results in editing accuracy and content consistency. Although DragGAN achieves better editing accuracy, it has limitations in content consistency and robustness in areas outside faces (*e.g.*, the headwear is distorted). The limitations of GAN-based DragGAN and UserControllableLT also exist in requiring alignment before editing, as shown in Fig. 8. It can be seen that if editing without alignment, the results of DragGAN will suffer from severe degradation. The alignment operation is not friendly to our editing goal, as it will change the original image content, *e.g.*, filtering out the background. In

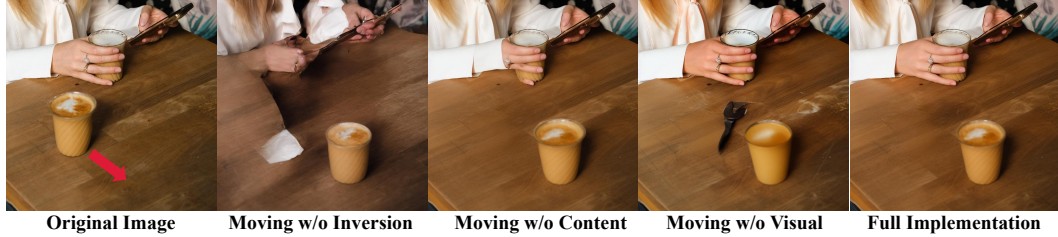

| Original Image | Moving w/o Inversion Prior | Moving w/o Content Consistency guidance $\mathcal{E}_{content}$ | Moving w/o Visual Cross-attention | Full Implementation |

Figure 9: Effectiveness of different components in our DragonDiffusion in the object moving task.

comparison, our method has promising editing accuracy, and the generation prior from SD enables better robustness and generalization for different content. In this task, our method also has better performance than DragDiffusion. More results are shown in the **appendix**.

**Other applications.** For object pasting, we compare our method with Paint-by-example Yang et al. (2023). For appearance replacing and object moving, we compare our method with Self-Guidance Epstein et al. (2024). The visual comparison in Fig. 7 shows that our method can achieve comparable performance to the training method (*i.e.*, Paint-by-example) in object pasting.

Compared to self-guidance, our method has better editing accuracy and content consistency. Due to the lack of consistency constraints, Self-Guidance produces some unexpected artifacts. Moreover, Self-Guidance has obvious deviation in complex scenes, due to the coarse correspondence between text and image features. More results are presented in Appendix.

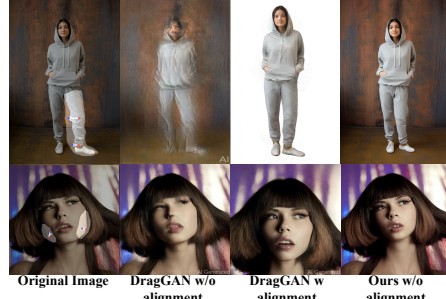

| Original Image | DragGAN w/o alignment | DragGAN w alignment | Ours w/o alignment |

Figure 8: Editing comparison between our DragonDiffusion and DragGAN Pan et al. (2023) on the unaligned body and face.

### 4.2 ABLATION STUDY

In this part, we demonstrate the effectiveness of some components in our DragonDiffusion, as shown in Fig. 9. We conduct the experiment on the object moving task. Specifically, **(1)** we verify the importance of the inversion prior by randomly initializing $\mathbf{z}_T$ instead of obtaining from DDIM inversion. As can be seen, the random $\mathbf{z}_T$ leads to a significant difference between the editing result and the original image. **(2)** We remove the content consistency guidance (*i.e.*, $\mathcal{E}_{content}$) in Eq. 7, which causes local distortion in the editing result, *e.g.*, the finger is twisted. **(3)** We remove the visual cross-attention. It can be seen that visual cross-attention plays an important role in maintaining the consistency between the edited object and the original object. Using a memory bank to provide $\mathbf{K}_t$ and $\mathbf{V}_t$ can greatly reduce the additional cost. In Appendix, we show an ablation study for memory bank. Therefore, these components work together on both edited and unedited content, forming the fine-grained image editing model DragonDiffusion, which does not require extra training or modules.

## 5 CONCLUSION

Despite the ability of existing large-scale text-to-image (T2I) diffusion models to generate high-quality images from detailed textual descriptions, they often lack the ability to precisely edit the generated or real images. In this paper, we aim to develop a drag-style and general image editing scheme based on the strong correspondence of intermediate image features in the pre-trained diffusion model. To this end, we model image editing as the change of feature correspondence and design energy functions to transform the editing operations into gradient guidance. Based on the gradient guidance strategy, we also propose multi-scale guidance to consider both semantic and geometric alignment. Moreover, a visual cross-attention is added based on a memory bank design, which can enhance the consistency between the original image and the editing result. Due to the reuse of intermediate information from the inversion process, this content consistency strategy almost has no additional cost. Extensive experiments demonstrate that our proposed DragonDiffusion can perform various image editing tasks, including object moving, resizing, appearance replacing, object pasting, and content dragging. At the same time, the complexity of our DragonDiffusion is acceptable, and it does not require extra model fine-tuning or additional modules.

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
