# APPEDIX-DRAGONDIFFUSION: ENABLING DRAG-STYLE MANIPULATION ON DIFFUSION MODELS

**Chong Mou**[1]    **Xintao Wang**[2]    **Jiechong Song**[1]    **Ying Shan**[2]    **Jian Zhang**[1*]

[1] School of Electronic and Computer Engineering, Shenzhen Graduate School, Peking University
[2] ARC Lab, Tencent PCG
`https://mc-e.github.io/project/DragonDiffusion/`

## A    APPENDIX

### A.1    ALGORITHM PIPELINE OF DRAGONDIFFUSION

To facilitate the understanding of our DragonDiffusion, we present the entire algorithm pipeline in Algorithm 1. Note that the text condition $\mathbf{c}$ has a minimal impact on the final result in our method, and only a brief description of the image is needed.

---

**Algorithm 1:** Proposed DragonDiffusion

**Require**:

text condition $\mathbf{c}$; UNet denoiser $\epsilon_{\boldsymbol{\theta}}$; pre-defined parameter $\bar{\alpha}_t$; image to be edited $\mathbf{x}_0$; the mask of $\mathbf{m}_{gen}$, $\mathbf{m}_{gud}$, and $\mathbf{m}_{share}$; the learning rate $\eta$; the number of gradient-guidance steps $n$.

**Initialization**:

(1) Compute latent $\mathbf{z}_0$ of the image to be edited:
  $\mathbf{z}_0 = Encoder(\mathbf{x}_0)$
(2) Select an editing task Ts $\in$ ['resizing&moving', 'dragging', 'pasting', 'replacing'];
(3) Compute latent $\mathbf{z}_0^{ref}$ of the reference image $\mathbf{x}_0^{ref}$:
  **if** Ts $\in$ ['resizing&moving', 'dragging'] **then** $\mathbf{z}_0^{ref} = \emptyset$ **else** $\mathbf{z}_0^{ref} = Encoder(\mathbf{x}_0^{ref})$
(4) Compute the inversion prior $\mathbf{z}_T^{gen}$ and build the memory bank:
  $\mathbf{z}_T^{gen}$, **Bank** $= DDIMInversion(\mathbf{z}_0, \mathbf{z}_0^{ref})$

**for** $t = T, \ldots, 1$ **do**

  **if** Ts $\in$ ['resizing&moving', 'dragging'] **then**

    $\mathbf{K}_t^{gud}$, $\mathbf{V}_t^{gud}$, $\mathbf{z}_t^{gud} = \mathbf{Bank}[t]$;

    $\mathbf{K}_t^{ref}$, $\mathbf{V}_t^{ref}$, $\mathbf{z}_t^{ref} = \emptyset$;

    extract $\mathbf{F}_t^{gen}$ and $\mathbf{F}_t^{gud}$ from $\mathbf{z}_t^{gen}$ and $\mathbf{z}_t^{gud}$ by $\epsilon_{\boldsymbol{\theta}}$;

    $\mathbf{K}_t$, $\mathbf{V}_t = \mathbf{K}_t^{gud}$, $\mathbf{V}_t^{gud}$;

  **else**

    $\mathbf{K}_t^{gud}$, $\mathbf{V}_t^{gud}$, $\mathbf{z}_t^{gud}$, $\mathbf{K}_t^{ref}$, $\mathbf{V}_t^{ref}$, $\mathbf{z}_t^{ref} = \mathbf{Bank}[t]$;

    extract $\mathbf{F}_t^{gen}$, $\mathbf{F}_t^{gud}$ and $\mathbf{F}_t^{ref}$ from $\mathbf{z}_t^{gen}$, $\mathbf{z}_t^{gud}$ and $\mathbf{z}_t^{ref}$ by $\epsilon_{\boldsymbol{\theta}}$;

    $\mathbf{K}_t$, $\mathbf{V}_t = \mathbf{K}_t^{gud}$©$\mathbf{K}_t^{ref}$, $\mathbf{V}_t^{gud}$©$\mathbf{V}_t^{ref}$;

  **end**

  $\hat{\epsilon}_t = \epsilon_{\boldsymbol{\theta}}(\mathbf{z}_t^{gen}, \mathbf{K}_t, \mathbf{V}_t, t, \mathbf{c})$;

  **if** $T - t < n$ **then**

    $\mathcal{E} = w_e \cdot \mathcal{E}_{edit} + w_c \cdot \mathcal{E}_{content} + w_o \cdot \mathcal{E}_{opt}$;

    $\hat{\epsilon}_t = \hat{\epsilon}_t + \eta \cdot \nabla_{\mathbf{z}_t}\mathcal{E}$;

  $\mathbf{z}_{t-1} = \sqrt{\bar{\alpha}_{t-1}}(\frac{\mathbf{z}_t - \sqrt{1-\bar{\alpha}_t}\hat{\epsilon}_t}{\sqrt{\bar{\alpha}_t}} + \sqrt{1-\bar{\alpha}_{t-1}}\hat{\epsilon}_t)$;

**end**

$\mathbf{x}_0 = Decoder(\mathbf{z}_0)$;

**Output:** $\mathbf{x}_0$

---

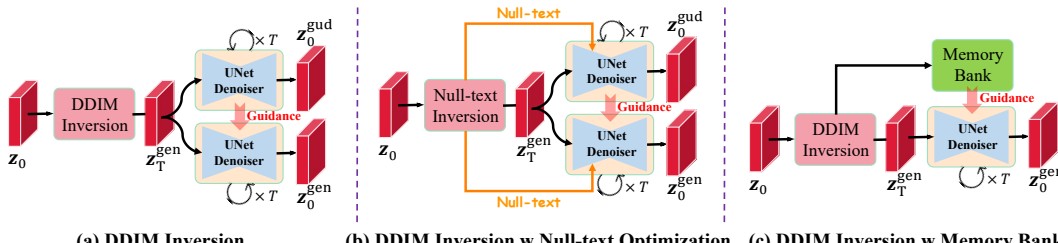

Figure 1: Different strategies for generating inversion prior (*i.e.*, $\mathbf{z}_T$) and guidance information (*i.e.*, $\mathbf{K}_t^{gud}, \mathbf{V}_t^{gud}$). (a) DDIM inversion + separate branch; (b) null-text inversion Mokady et al. (2023) + separate branch; (c) our memory bank design.

## A.2    EFFICIENCY OF THE MEMORY BANK DESIGN

In this paper, we designed a memory bank to store intermediate information during the inversion process, which is used to provide guidance for image editing. To verify its effectiveness, we compared it with methods having the same function, as shown in Fig. 1. Specifically, (a) guidance information is generated by a separate generation branch from $\mathbf{z}_T$; (b) null-text optimization is added based on method (a); (c) using our designed memory bank strategy. The editing quality of different methods is presented in Fig. 3. It can be seen that extracting guidance information from $\mathbf{z}_T$ using a separate branch can lead to deviations. This is due to the approximation bias in DDIM inversion. Although incorporating null-text optimization can yield more accurate results, it comes with higher time complexity. Our method tactfully utilizes a memory bank to store intermediate information during the inversion process, achieving accurate results while maintaining a time complexity of only 3.62 seconds.

## A.3    THE EFFECTIVENESS OF $\mathcal{S}_{global}$

In our method, $\mathcal{S}_{global}$ is designed to control the global appearance of a specific region. It exists in two parts: (1) in the object moving task, it is used to control the appearance consistency between the area where the object originally located and a reference area (*i.e.*, reference-based inpainting); (2) in the appearance replacing task, it is used to control the appearance consistency between the edited object and a reference object. In Fig. 4, we demonstrate the role of $\mathcal{S}_{global}$. It can be seen that without using $\mathcal{S}_{global}$, the generation of the inpainting area in the object moving task will lack guidance, resulting in distortion. In the appearance replacing task, although there is visual cross-attention (*i.e.*, Eq. **??**) providing content reference, removing $\mathcal{S}_{global}$ will cause the editing area to lose guidance on whether to use information in the original image or reference image, leading to mixed content in the result. Therefore, $\mathcal{S}_{global}$ plays an important role in appearance control.

## A.4    ADDITIONAL ABLATION STUDY FOR HYPERPARAMETERS

**Loss weight**. Our energy function $\mathcal{E}$ for image editing is mainly composed of editing guidance $\mathcal{E}_{edit}$ and content consistency guidance $\mathcal{E}_{content}$, with their weights being $w_e$ and $w_c$, respectively. We conducted ablation experiments on the effects of $w_e$ and $w_c$ in Fig. 5. It can be seen that both these two weights can strengthen the importance of their respective energy functions by increasing their weights. However, there is an interaction between them. The interaction here is mainly reflected in the influence of $\mathcal{E}_{edit}$ on $\mathcal{E}_{content}$. In comparison, the increase of $\mathcal{E}_{content}$ does not have a significant impact on $\mathcal{E}_{content}$. Therefore, in our design, we let $w_c$ be slightly larger than $w_e$ to achieve the editing goal while providing better content consistency.

**Time step**. Our inference process adopts 50-step DDIM sampling, but the effect of adding gradient guidance at different time steps is different. The results in Fig. 6 show that it is important to add guiding information in the early stage of sampling. The later the sampling, the weaker the role of guiding information. In addition, adding guiding information in more sampling steps helps improve

---
*Corresponding author

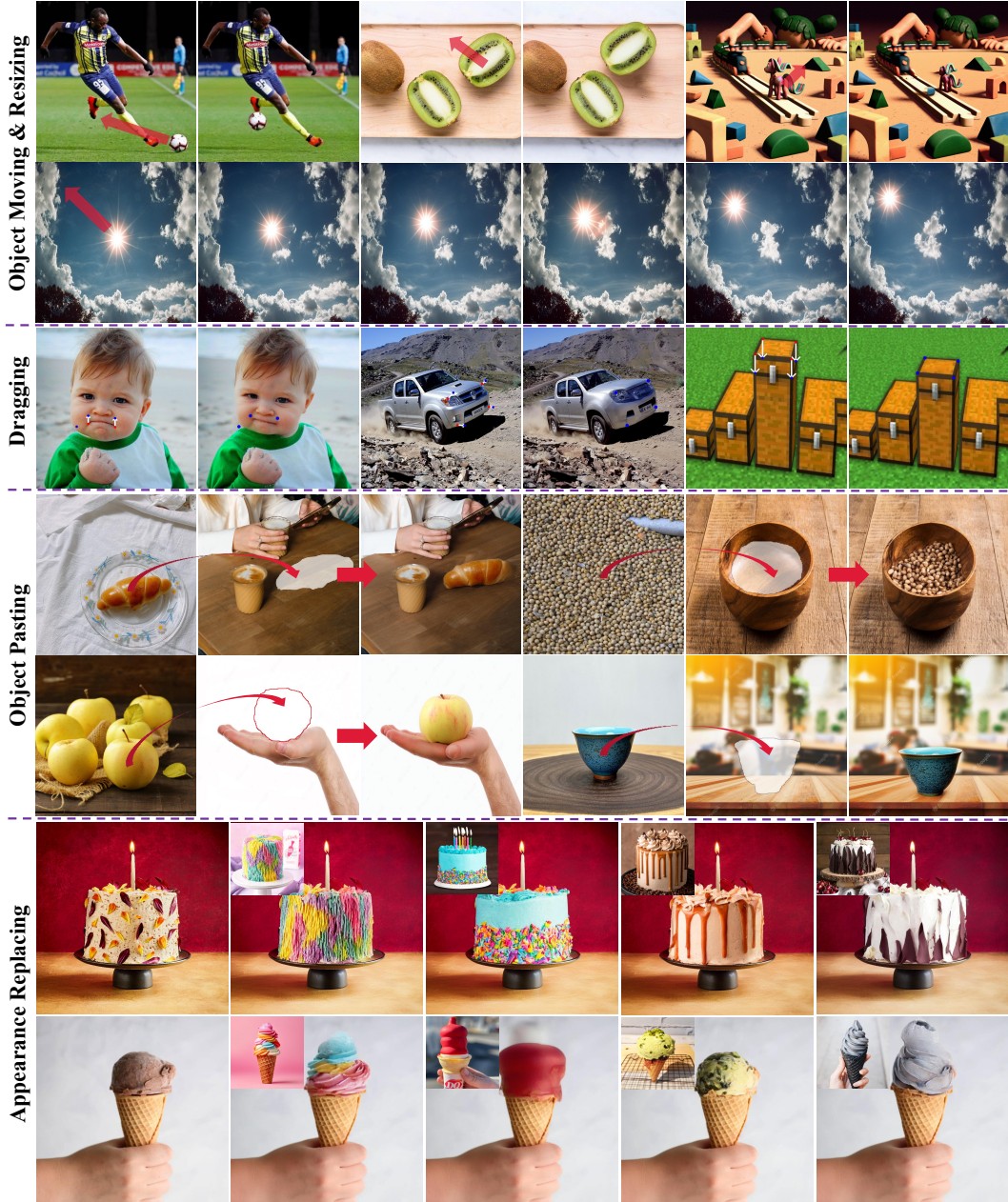

Figure 2: More results of DragonDiffusion on different applications.

the quality of texture details. After balancing the performance and computational complexity, we choose to add gradient guidance in the first 30 steps.

**Text prompt**. Our method is based on pre-trained StableDiffusion Rombach et al. (2022), so there is text input. We show the impact of text on the editing results in Fig 8. It can be seen that the editing results produced by different texts are close and meet the editing requirements. The editing objectives can be achieved even under completely unrelated texts (*e.g.*, *"a boy plays with a dog"*) or texts that conflict with the image (*e.g.*, *"no cup, no table, no woman"*).

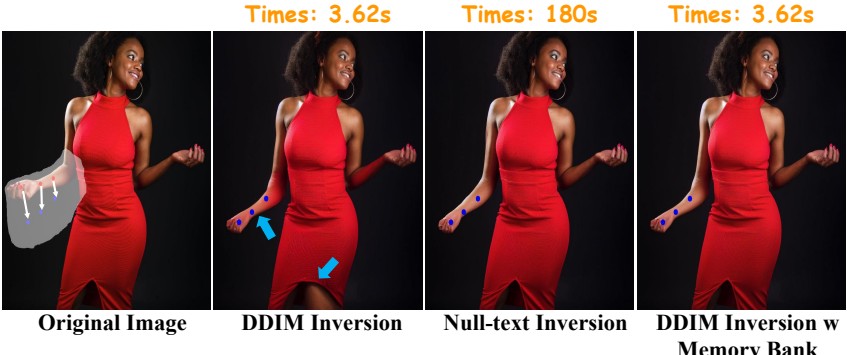

Figure 3: The editing quality of different guidance strategies.

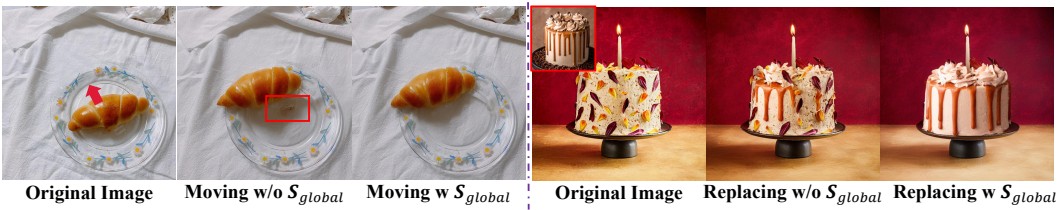

Figure 4: Visualization of the effectiveness of $\mathcal{S}_{global}$.

### A.5 VISUALIZATION OF THE GRADIENT GUIDANCE

In Fig. 7, we visualize the gradient maps in the object moving task at different time steps. The first row and the second row are produced by $\mathcal{E}_{edit}$ and $\mathcal{E}_{content}$ respectively, showing a gradually converging process. Specifically, as the sampling proceeds, the activation range of the gradient map narrows down, gradually converging to their respective editing areas.

### A.6 MORE RESULTS OF DRAGONDIFFUSION ON DIFFERENT APPLICATIONS

In this part, we present more visual results of our DragonDiffusion in different applications, as shown in Fig. 2. The first and second rows show the visualization of our object moving performance. It can be seen that our method has attractive object moving performance and good content consistency even in complex scenarios. The continuous moving editing presents attractive editing stability. The third row demonstrates that our method can perform natural point-drag editing of image content in different scenarios with several points. The fourth and fifth rows show the performance of our method in cross-image object pasting tasks. It can be seen that our method can fine-tune an object in one image and then naturally paste it onto another image. The last two rows demonstrate the performance of our method in object appearance replacing. It can be seen that our DragonDiffusion not only has good editing quality on small objects (*e.g.*, ice-cream) but also performs well in replacing the appearance of large objects (*e.g.*, cakes). Therefore, without any training and additional modules, our DragonDiffusion performs well in various image editing tasks.

### A.7 MORE QUALITATIVE COMPARISONS BETWEEN OUR DRAGONDIFFUSION AND OTHER METHODS ON CONTENT DRAGGING

In this part, we demonstrate more qualitative comparisons between our DragonDiffusion and other methods on more categories. Fig. 13 shows the comparison of drag editing on dogs. Fig. 14 shows the comparison of drag editing on horses. Fig. 15 shows the comparison of drag editing on cars. Fig. 16 shows the comparison of drag editing on churches and elephants. Fig. 17 shows the comparison of drag editing on face manipulation. In these comparisons, DragGAN Pan et al. (2023) requires switching between different models for different categories. Our method and DragDiffusion Shi et al. (2023) benefit from the powerful generalization capabilities of SD Rombach et al. (2022), enabling a single model to address image editing across different categories. These visu-

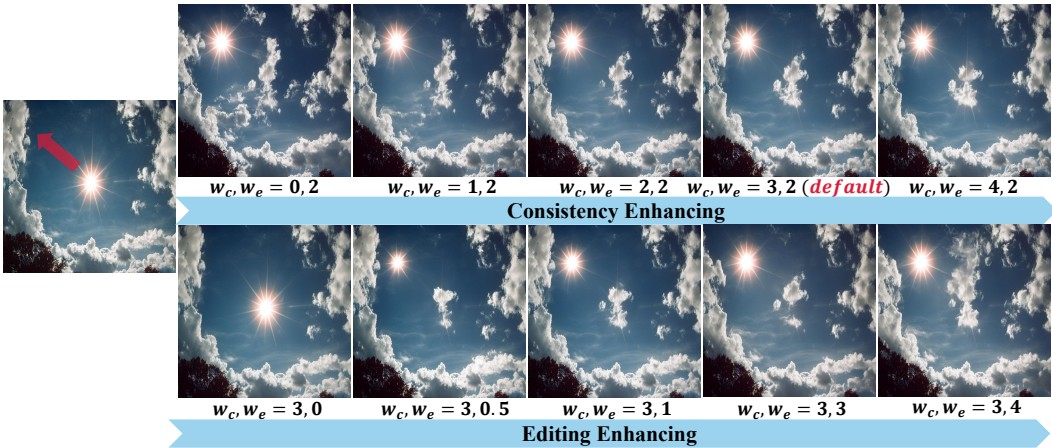

Figure 5: Visualization of the trade-off between editing gradient weight $w_e$ and content consistency gradient weight $w_c$.

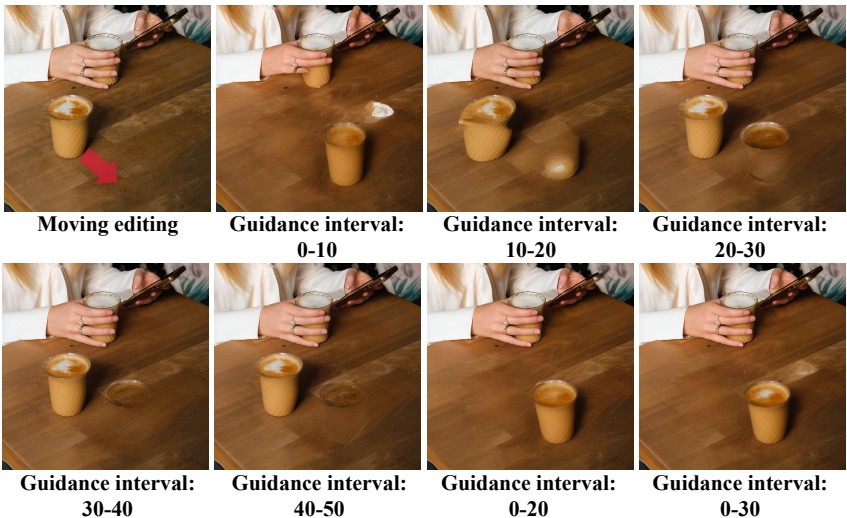

Figure 6: Editing results generated by providing guidance at different time steps.

alization results show that our method can produce better consistency with original images. At the same time, our method well balances the editing accuracy and generation quality.

To further demonstrate the capabilities of our method, we show the editing quality of DragGAN, DragDiffusion, and our method in challenging scenarios in Fig. 9. As can be seen, test samples of the car, face, and elephant deviate from the training data of their corresponding StyleGANs. In such cases, the editing quality of DragGAN deteriorates severely. Our method, based on the SD, can handle the editing of general images more effectively. Compared to DragDiffusion, our method performs better in content consistency and editing accuracy in these challenging cases.

## A.8 MORE COMPARISONS BETWEEN OUR DRAGONDIFFUSION AND OTHER METHODS ON OBJECT PASTING, OBJECT MOVING, AND APPEARANCE REPLACING

In Fig. 10, we present more visual comparison between our method and: (1) Paint-by-example Yang et al. (2023) on object pasting; (2) Self-Guidance Epstein et al. (2023) on object moving and appearance replacing. The results show that our DragonDiffusion can better preserve the object identity in object pasting. Meanwhile, compared to Paint-by-example, our method does not require training. In object moving and appearance replacing tasks, our method achieves higher editing accuracy and content consistency than Self-Guidance. Quantifying these tasks is challenging. We compute the

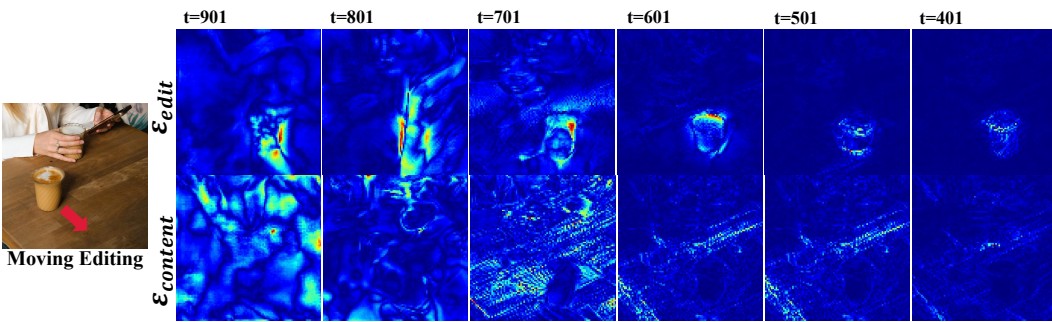

Figure 7: Visualization of the editing gradient $\mathcal{E}_{edit}$ and content consistency gradient guidance at different diffusion time steps.

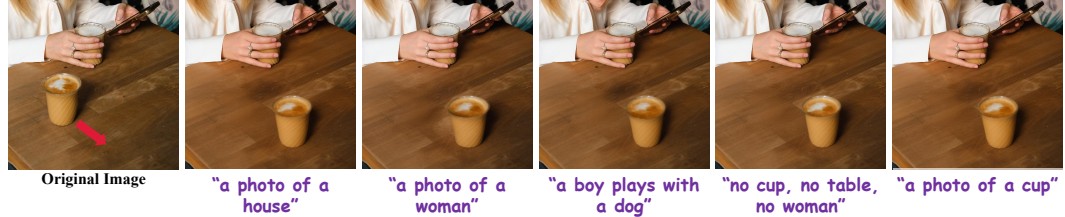

Figure 8: Visualization of the effectiveness of text in our image editing process. We can find that the text has little impact on the results of our image editing method.

CLIP distance between the edited results and the target description. We select 8 editing samples for each task. The results in Tab. 1 demonstrate the promising performance of our method.

## A.9 FURTHER DISCUSSION WITH TEXT-GUIDED IMAGE EDITING

In this chapter, we further discuss the differences between text-guided image editing methods (*e.g.*, Null-text inversion Mokady et al. (2023), InstructPix2Pix Brooks et al. (2023), and Self-Guidance Epstein et al. (2023)) and our proposed DragonDiffusion based on image feature correspondence. The premise of these text-guided image editing methods to complete the editing is that the text and the content to be edited have a local correspondence. However, this correspondence is weak in complex scenarios. For example, as shown in Fig. 11, these methods will fail when editing a certain object in multi-object images. Moreover, Null-text inversion and InstructPix2Pix cannot select reference images. In comparison, our method can accurately select the editing content and set the editing target, resulting in better editing results.

## A.10 USER STUDY

To further compare with DragGAN Pan et al. (2023) and DragDiffusion Shi et al. (2023), we design a user study, which includes three evaluation aspects: generation quality, editing accuracy, and content consistency. The test samples involve various categories including dog, horse, car, elephant, church, and face. We allow 20 volunteers to choose the best-performing method in each of the 16 groups of images and then compile the votes in Fig. 18. As can be seen, our method has better subjective performance in these three aspects.

## A.11 DEMO VIDEO

A demo video is attached to the supplementary materials.

## A.12 OUR LIMITATIONS

Although our method can perform various image editing tasks without training, there are some limitations. (1) There is still room for further optimization in inference time complexity. A possible

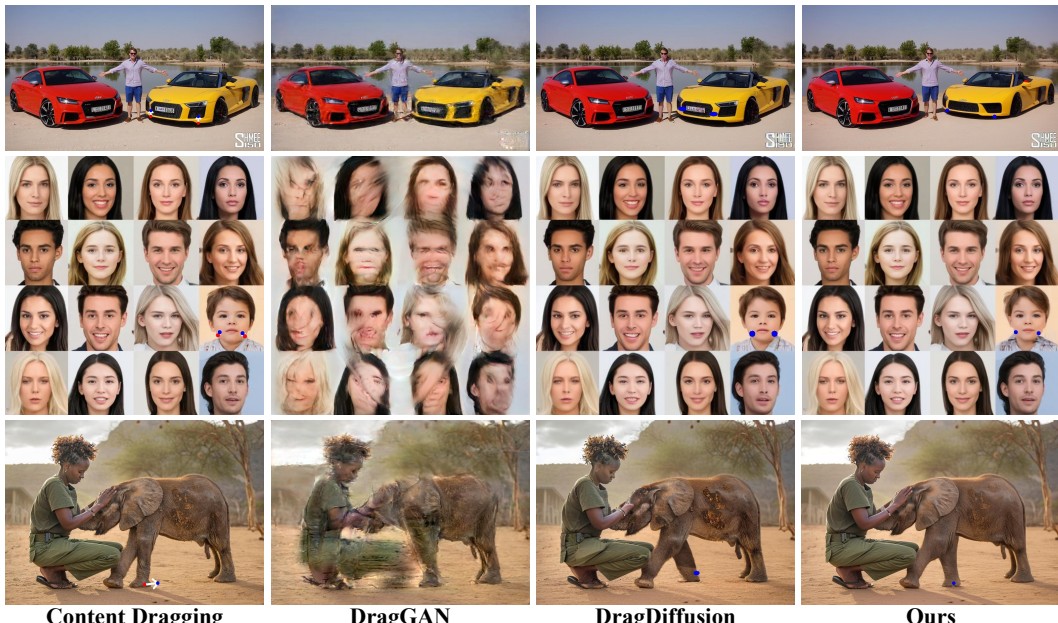

**Content Dragging**     **DragGAN**     **DragDiffusion**     **Ours**

Figure 9: Qualitative comparison between DragGAN Pan et al. (2023), DragDiffusion Shi et al. (2023), and our DragonDiffusion in more complex scenarios.

Table 1: Quantitative evaluation on object pasting, object moving, and appearance replacing. The result is calculated by CLIP distance between editing results and target descriptions.

|                | Object pasting | Object moving | Appearance replacing |
|----------------|:--------------:|:-------------:|:--------------------:|
| Pain-by-example | 0.253          | -             | -                    |
| Self-Guidance   | -              | 0.258         | 0.247                |
| Ours            | 0.262          | 0.279         | 0.268                |

scheme is to reuse gradient maps of adjacent time steps, further reducing the complexity of gradient calculation. (2) Since our method does not require training, some hyperparameters involved in the editing process may affect the editing results, as discussed in Sec. A.4. In future work, we plan to explore plug-and-play auxiliary modules to improve the editing performance. It is acceptable if the weaknesses in existing editing can be addressed through a single training session.

### A.13   A FURTHER DISCUSSION OF EDITING GUIDANCE IN DRAGDIFFUSION AND DRAGONDIFFUSION

Our method differs from DragDiffusion in editing guidance. DragDiffusion selects a latent $\mathbf{z}_t$ in the diffusion process as a learnable parameter and optimizes $\mathbf{z}_t$ through multiple iterations. However, this optimization method only considers the editing target and ignores the impact on the diffusion process, *i.e.*, it cannot guarantee that the optimized $\mathbf{z}_t$ conforms to the current time step t. In contrast, our method is built on score-based gradient guidance to solve $p(\mathbf{z}_{t-1}|\mathbf{z}_t, y)$, where $y$ is the editing target. Dhariwal & Nichol (2021) has proved that $p(\mathbf{z}_{t-1}|\mathbf{z}_t, y) \sim \mathcal{N}(\mu + \eta g, \Sigma)$, where $\eta g$ is the gradient guidance produce from an energy function $p(y|\mathbf{z}_t)$. Therefore, our guidance method is compatible with the diffusion process. The energy function (*i.e.*, feature correspondence Tang et al. (2023)) in our method is also proven to be accurate and robust in measuring the feature correspondence.

We show the editing results of these two guidance methods in Fig. 12. As can be seen, without the constraint of LORA, the latent optimization (under 68 points) of DragDiffusion causes $\mathbf{z}_t$ to deviate from the sampling distribution, resulting in unreasonable results. In comparison, our method still produces reasonable results after removing visual cross-attention. Only the content consistency is

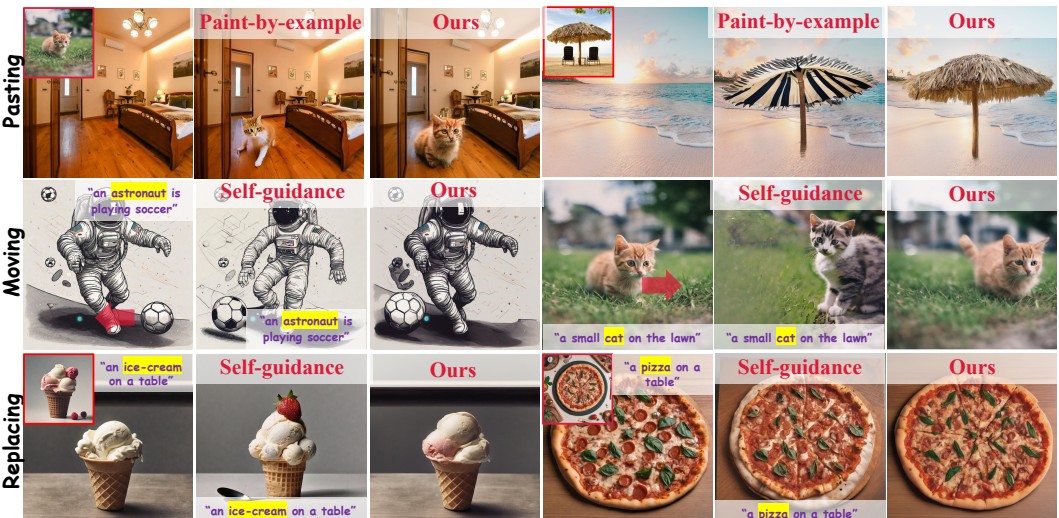

Figure 10: More qualitative comparison between our DragonDiffusion and other methods on object pasting (first row), object moving (second row), and appearance replacing (third row).

not well preserved. Therefore, our guidance method is more friendly to diffusion models and can generate results with higher quality.

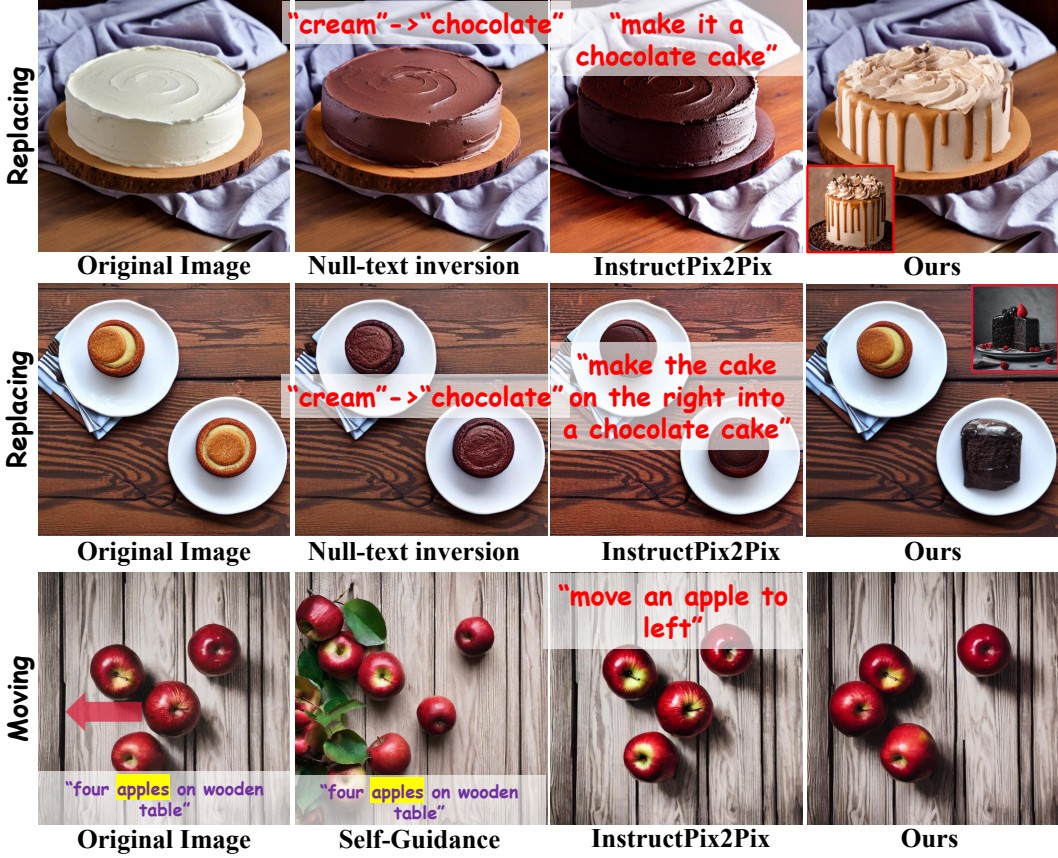

Figure 11: Editing comparison between our DragonDiffusion and text-guided image editing methods, *e.g.*, Null-text inversion Mokady et al. (2023), InstructPix2Pix Brooks et al. (2023), and self-guidance. Their editing is coarse, and it will fail in multi-object situations.

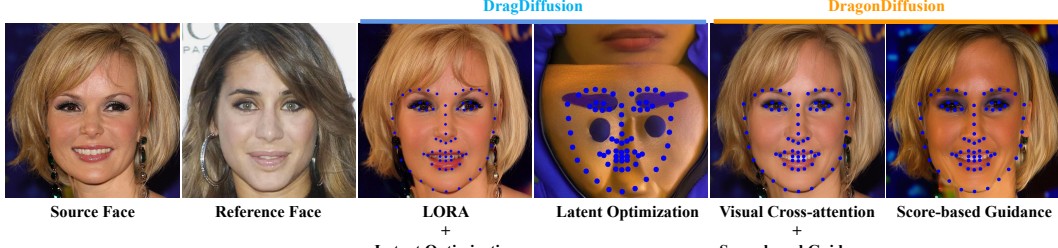

Figure 12: Editing comparison between the guidance in DragDiffusion (latent optimization) and our DragonDiffusion (score-based guidance).

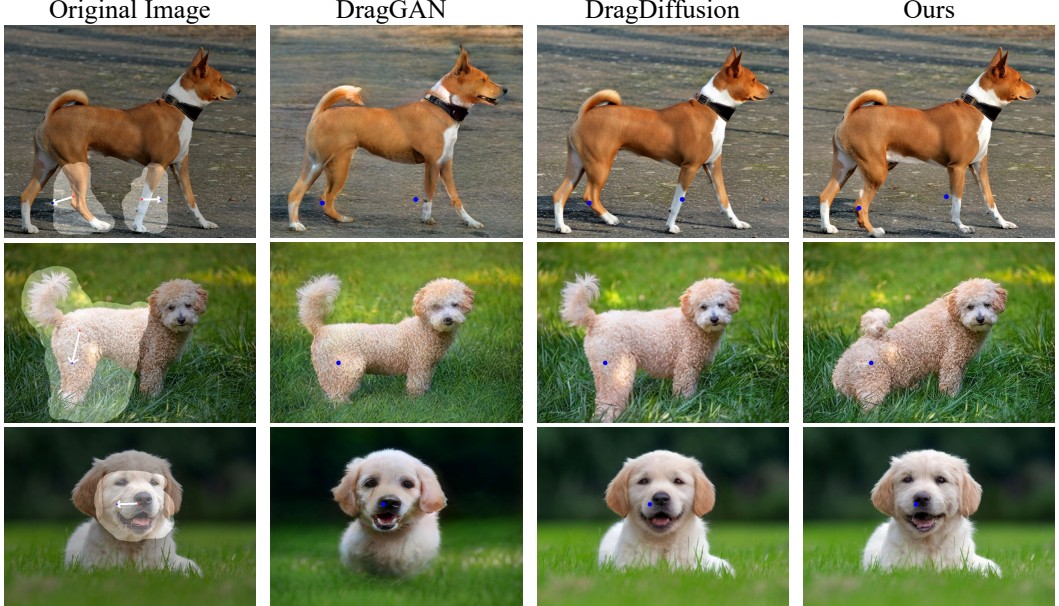

Figure 13: More qualitative comparison between our DragonDiffusion and other methods on the dog dragging. It can be seen that DragGAN Pan et al. (2023) is limited in generation quality and content consistency due to the capabilities of GAN models. DragDiffusion Shi et al. (2023) experiences an accuracy decline when dealing with larger editing drags, such as changing the posture of the dog's body. In comparison, our method has promising performance in these aspects.

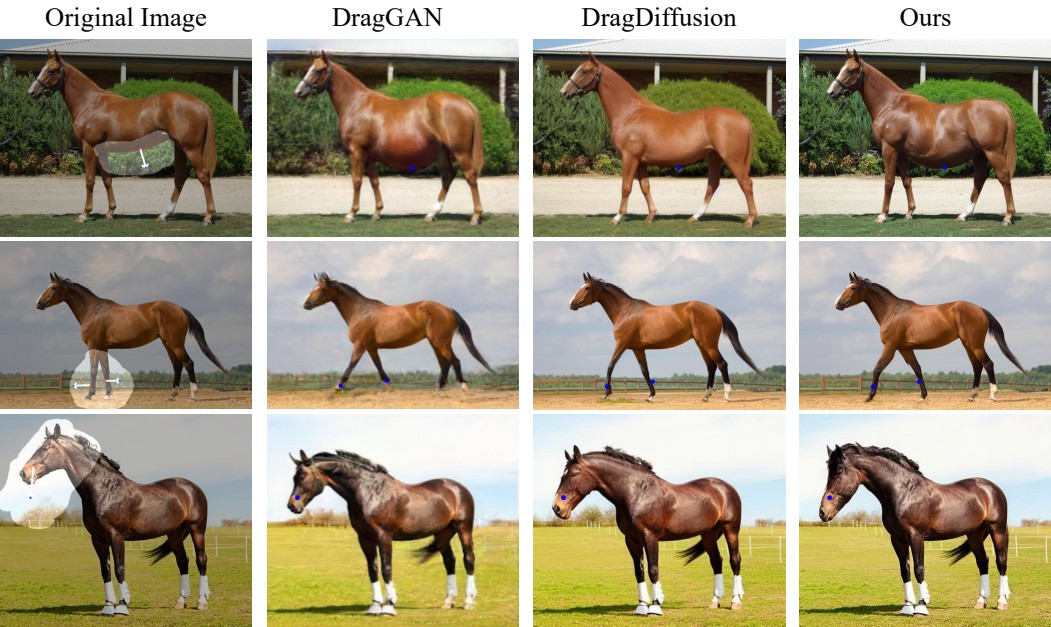

Figure 14: More qualitative comparison between our DragonDiffusion and other methods on the horse dragging.

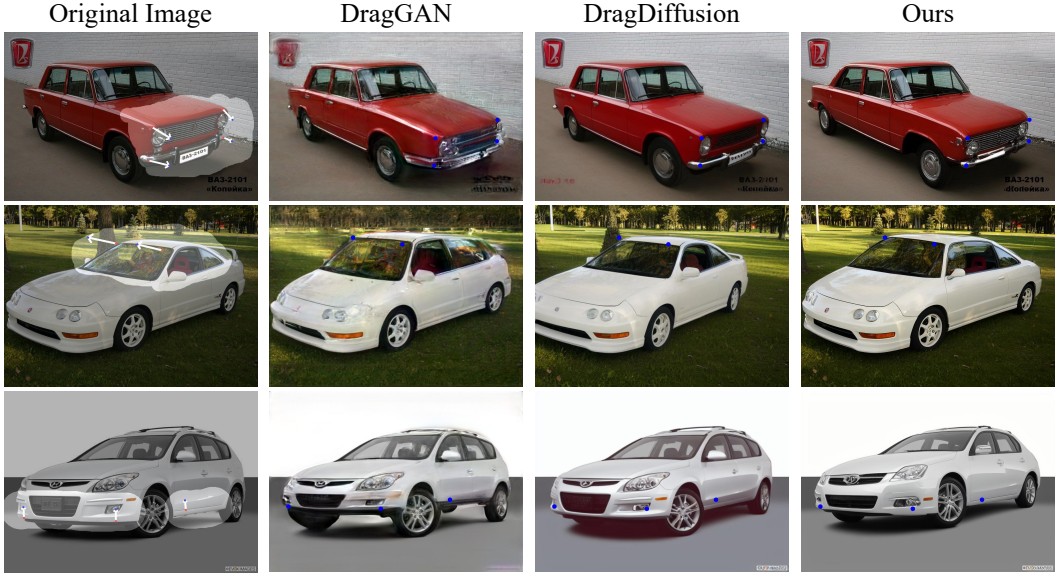

Figure 15: More qualitative comparison between our DragonDiffusion and other methods on the car dragging.

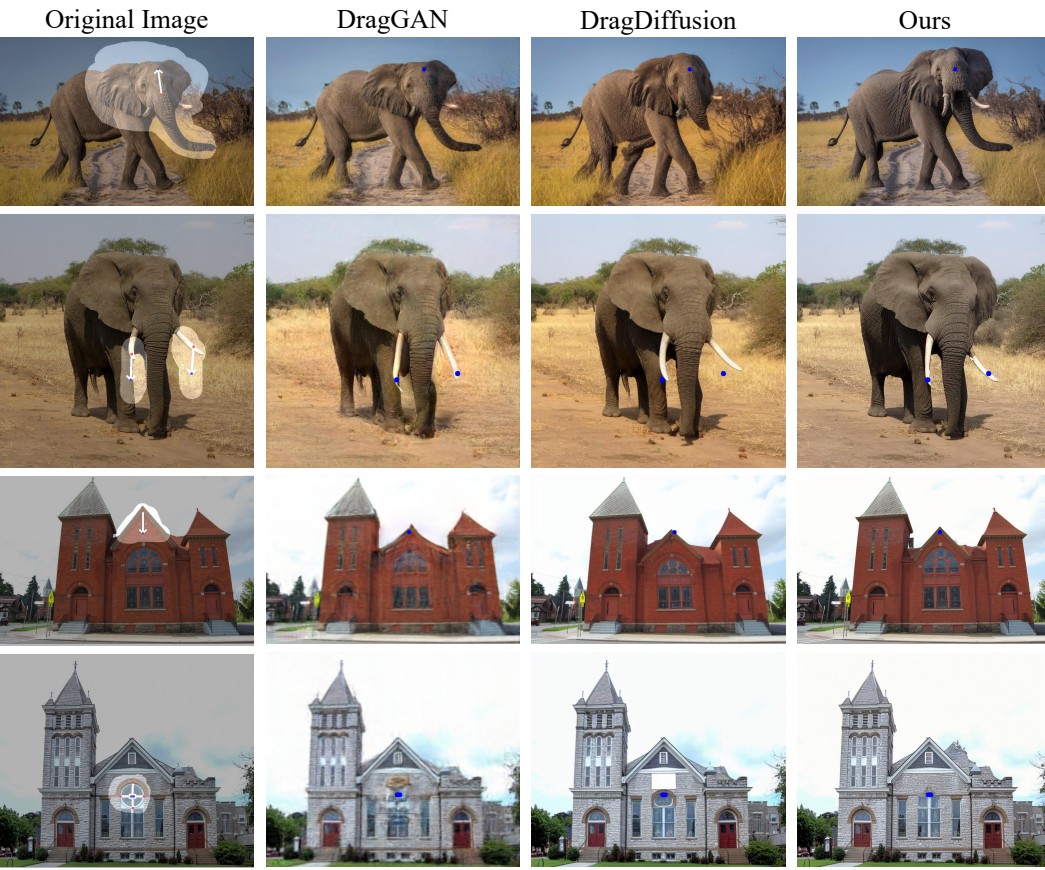

Figure 16: More qualitative comparison between our DragonDiffusion and other methods on the church and elephant dragging.

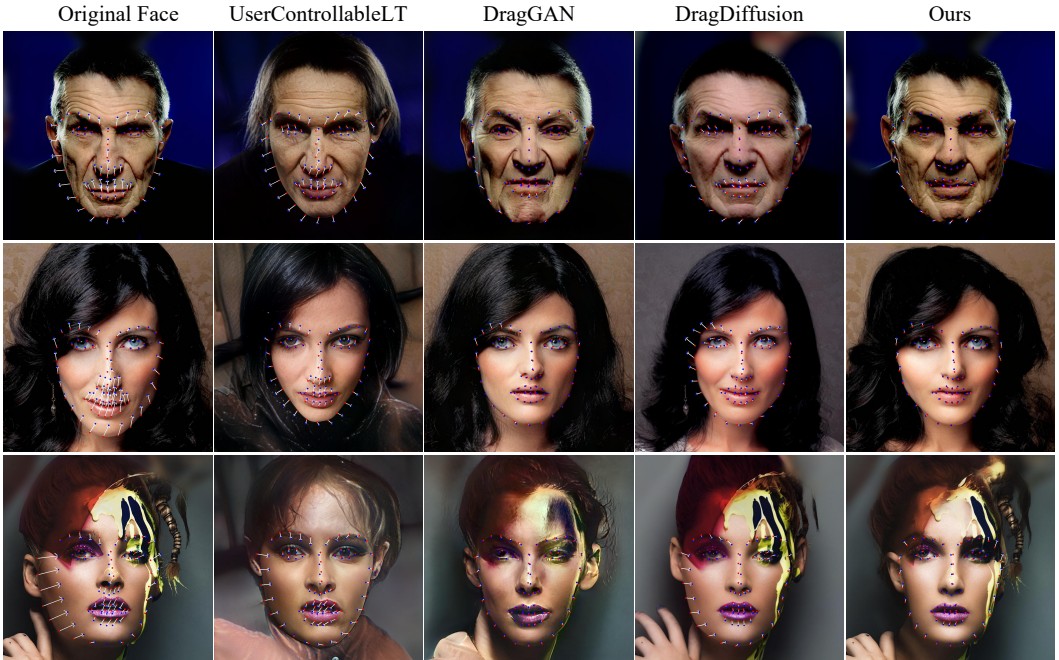

Figure 17: More qualitative comparison between our DragonDiffusion and other methods on the face dragging.

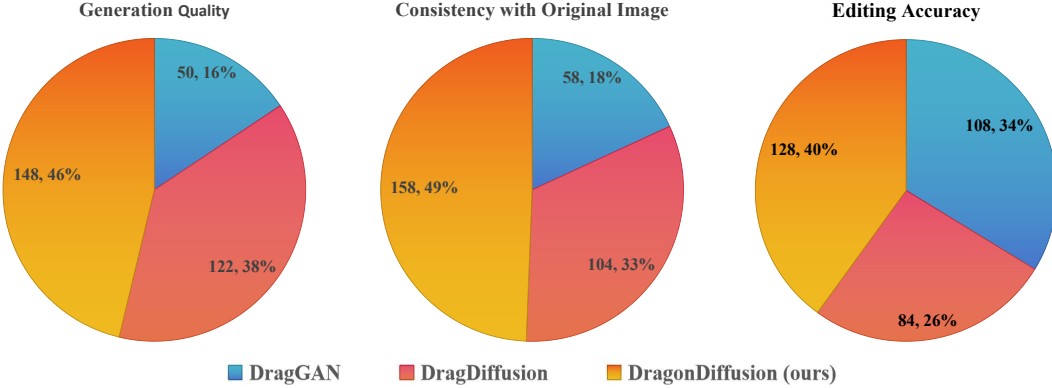

Figure 18: User study of DragGAN Pan et al. (2023), DragDiffusion Shi et al. (2023), and our DragonDiffusion. The experiment is conducted on various categories including dog, horse, car, elephant, church and face.