# OpenReview forum: "DragonDiffusion: Enabling Drag-style Manipulation on Diffusion Models"
_ICLR.cc/2024/Conference — ICLR 2024 spotlight_

### Official Review · Reviewer_8eDU · 2023-10-26

**Soundness:** 2 fair
**Presentation:** 3 good
**Contribution:** 3 good
**Rating:** 6
**Confidence:** 4

**Summary:**

This paper presents DragonDiffusion - a novel framework to enable drag-style image editing with diffusion models.
To this end, new techniques are presented, including (1) using an energy function to guide the editing and (2) a memory bank for editing consistency.
Qualitative and quantitative experiments show the merits of DragonDiffusion.

**Strengths:**

- The idea of enabling more precise and interactive image editing with diffusion models is an attractive topic. I believe that this work will attract both academic and community interest.
- Sufficient experiments have been conducted to compare with DragGAN-related methods, showing the merits of the methods (e.g., DragGAN can not edit based on a reference image).
- The ablation study demonstrates the effectiveness of the framework design.
- Generally, the paper is easy to follow.

**Weaknesses:**

1. My primary concern about this paper is **whether ICLR is a suitable venue**.
I believe this paper would be more fitting for a Computer Vision conference (e.g., CVPR, ICCV, ECCV, SIGGRAPH).
While I don't intend to downplay the contribution of this paper (in fact, I appreciate it), I find it challenging to identify a precise description for this paper within the context of ICLR.
Perhaps, "representation learning for application in Computer Vision", but given that there is no "representation learning" happening, I am not sure. Thus, my initial rating is "marginally below the acceptance threshold".

1. My other complaints are mainly about *writing* (but it is not the main reasons for my decision). Authors can use it to improve paper' clarity.
- In Abstract, "... they often lack the ability to precisely edit the generated or real images." -> I think this should tone down to "interactively" as "precisely" might have a broad meaning (e.g., precise in terms of pose, shape, etc.). In a broader meaning of precision, I see existing works can also achieve "precise" image editing (e.g., ControlNet [1]).
- Introduction, first paragraph: (Similar to above) While I see DragonDiffusion has clearly advanced in interactive image editing, I think it'd be more comprehensive to mention seminal works aiming to perform more precise image editing (e.g., [1-4]... you name it). Alternatively, authors can briefly discuss these works in Section 2.3.
- Section 2.3, "InstructPix2Pix retrain diffusion models.." -> "InstructPix2Pix finetunes diffusion models..."
- Section 2.3, "However, text-guided image editing is coarse." Could you add a sentence explaining why "coarse" is a bad thing?
- Section 2 though Section 3.1 all use $x_{T}$, then suddenly Section 3.2 uses $z_{T}$. As far as I understand, the authors intend to use Latent Diffusion (Stable Diffusion), which is $z_{T}$. Then, could authors revise Section 2.1 (and other related parts in Section 2-3), so it is made sure that we have mentioned $z_{T}$ before?

**Reference**:
[1] Zhang et al., Adding Conditional Control to Text-to-Image Diffusion Models, ICCV, 2023.
[2] Couairon et al., DiffEdit: Diffusion-based semantic image editing with mask guidance, ICLR, 2023.
[3] Nguyen et al., Visual Instruction Inversion: Image Editing via Visual Prompting, NeurIPS, 2023.
[4] Epstein et al., Diffusion Self-Guidance for Controllable Image Generation, NeurIPS, 2023.

**Questions:**

As both DragonDiffusion and Self-Guidance [4] use (1) an energy function and (2) modify the attention layer to perform edits, could the author elaborate further on the differences between them? I also think it would be great if the author could compare them to Self-Guidance (as they can also resize objects, move objects, etc.).

---

> ### Author Response · Authors · 2023-11-14
>
> Thank you for your careful review and constructive suggestions. We have modified our paper to rectify your concerns. The changes relevant to your concerns are marked in orange (color will be removed in the final version).
>
> # W1. Whether ICLR is a suitable venue
>
> In our method design, we study the feature representation learned by the pre-trained diffusion model to design energy functions. During the editing process, we inject the gradient guidance generated by the energy function into the discrete latent representation of the pre-trained diffusion model, achieving a general image editing framework. Therefore, our method is also a study of the feature representation and latent representation of diffusion models.
>
> In addition, we check the Subject Areas of ICLR, which include applications in vision, audio, speech, language, music, etc. Therefore, we believe that our work is suitable for ICLR.
>
> # W2. Writing
>
> Thank you for these constructive suggestions. We have reviewed the relevant sections and made the necessary modifications.
>
> # Q1. Compare with self-guidance
>
> Yes, comparing with self-guidance is necessary. Since self-guidance was not open-sourced before the ICLR submission, our original paper did not include a performance comparison with it. Our main difference lies in modeling editing operations. Self-guidance calculates the similarity between the text (corresponding to the object to be edited) and image features, selects regions with high similarity as editing areas, and then applies editing operations. Therefore, the attention in self-guidance is used to locate editing operations. However, this text-guided image editing is coarse. For example: (1) Text can only locate content at the object level, and more accurate local content cannot be expressed in text, making it impossible to be edited with self-guidance. (2) In multi-object scenes, a single text corresponds to multiple objects, making it impossible to use self-guidance to edit a specific object. (3) Text can not provide effective content consistency, leading to deviations between the editing results and the original image. In contrast, our method models editing operations based on the correspondence of intermediate features in the pre-trained diffusion model, which can achieve pixel-to-pixel correspondence, and the editing area is manually defined. Therefore, our method can achieve fine-grained image editing, such as content dragging, which is not supported by self-guidance. In our method, attention is used to inject content consistency guidance, resulting in better consistency between the edited results and the original image.
>
> In Sec.4.1 of the updated version, we add the comparison between our method against Self-Guidance in object moving and appearance replacing. We add more comparisons in Appendix.8. These results demonstrate that our method has higher editing accuracy and content consistency.

---

> > ### Author Response · Authors · 2023-11-21
> > **Looking forward to further feedbacks**
> >
> > Dear Reviewer 8eDU,
> >
> > Thank you again for your great efforts and valuable suggestions. We have carefully addressed your concerns. We hope you are satisfied with our response. As the discussion phase is about to close, we are looking forward to hearing from you about any further feedback. We will be happy to clarify further concerns (if any).
> >
> > Best,
> >
> > Authors

---

> > > ### Comment · Reviewer_8eDU · 2023-11-22
> > >
> > > Thank you for continuously updating the submission.
> > > To be honest, except from the writing, I do not have any other complaints.
> > > My major complaint is about whether this submission is suitable for ICLR or not.
> > > Thus, my rating is positive, but I'll discuss this with Area Chair.
> > >
> > > Also: Please update the manuscript accordingly on this point: "Section 2 though Section 3.1 all use $x_{T}$, then suddenly Section 3.2 uses $z_{T}$. As far as I understand, the authors intend to use Latent Diffusion (Stable Diffusion), which is $z_{T}$. Then, could authors revise Section 2.1 (and other related parts in Section 2-3), so it is made sure that we have mentioned $z_{T}$ before?"
> > >
> > > Good luck!

---

> > > > ### Author Response · Authors · 2023-11-22
> > > >
> > > > Thank you for your efforts and positive rating. In Sec. 3.2, we update a description that SD is a latent diffusion model (LDM), as well as the relationship between $x_t$ and $z_t$.

---

> ### Author Response · Authors · 2023-11-16
>
> Dear reviewer, thanks a lot for your previous constructive comments. We would like to know if our revisions have addressed your concerns? We welcome any discussions and suggestions that will help us further improve this paper.

---

> > ### Comment · Reviewer_8eDU · 2023-11-19
> >
> > Thank you for updating the manuscript!
> > I went through your comparison with Self-Guidance, can you please provide the text description that you use for each images? (Fig. 7, Fig. 18, Fig. 19).

---

> > > ### Author Response · Authors · 2023-11-19
> > >
> > > Thank you for the useful suggestion:) Self-Guidance is a text-guided image editing method, and it is indeed necessary to display the text during its editing process. We have updated the paper (Fig. 7, Fig. 18, Fig. 19) to include the text used in Self-Guidance editing. The words corresponding to the edited objects are marked with yellow boxes. We also find it important to keep the text as simple as possible for Self-Guidance. Otherwise, it will affect the correspondence between the word and the editing object. In comparison, text condition has a minimal impact on our method.

---

> > > ### Author Response · Authors · 2023-11-21
> > > **The impact of text input for our method**
> > >
> > > To further solve this concern, we add an ablation experiment for text in Appendix 4. It shows that the editing results produced by different texts are close and meet the editing requirements. The editing objectives (moving a cup on the table) can be achieved even under completely unrelated text (e.g., $"a\ boy\ plays\ with\ a\ dog"$) or text that conflicts with the image (e.g., $"no\ cup,\ no\ table,\ no\ woman"$).

---

### Official Review · Reviewer_dwdi · 2023-10-28

**Soundness:** 4 excellent
**Presentation:** 3 good
**Contribution:** 2 fair
**Rating:** 6
**Confidence:** 4

**Summary:**

In this paper, the authors propose a method of drag-style image manipulation with Diffusion Model. They addressed this through the guidance based on feature matching and also conducted comparisons with previous studies such as DragGAN and DragDiffusion. The effects of using layers of various scales were analyzed, and attempts were made to preserve the content of the original image using visual cross attention. In addition, various applications such as object moving, object resizing, appearance replacing, and object pasting were also demonstrated.

**Strengths:**

- The training time is short and FID score is better compared to DragGAN and DragDiffusion
- Many applications are conducted like object moving, object resizing, appearance replacing, and object pasting

**Weaknesses:**

- The problem targeted by this paper is not clear. Therefore, it is unclear why diffusion feature matching, drag-style editing, memory bank, and visual cross-attention strategy were introduced, giving an incremental feel.
- If the paper is focused on the problem of drag-style image manipulation, more experimental results should be presented. For example, it is unclear why the FID score is higher compared to DragGAN. There is no related ablation study for that part.
- Despite the introduction of the visual cross-attention strategy, it feels like that the identity or content of the image is not sufficiently preserved.
- The choices of hyper-parameters seems heuristic. The experiment from multiple combination of hyper-parameter set could be helpful to address this issue.

**Questions:**

- Just as selecting the feature layer or combining information from the layer is important, I know that at which diffusion time the guidance is given also has a significant impact. Were there any related experiments on this?

---

> ### Author Response · Authors · 2023-11-14
>
> Thank you for your careful review and constructive suggestions. We have modified our paper to rectify your concerns. The changes relevant to your concerns are marked in purple (color will be removed in the final version).
>
> # W1. The problems targeted in this paper.
>
> - drag-style editing is the target task
>
> In this paper, we want to build a general framework to complete drag-style image editing, e.g., object moving, resizing, pasting, content dragging, and appearance replacing. This has not been achieved before.
>
> - feature matching builds the energy guidance as an editing tool without training
>
> In this paper, we choose energy guidance in diffusion models to achieve image editing. Energy guidance is the second term in Eq.2 in our original paper, which combines the generation target y with the diffusion process as q(y|$x_t$). Therefore, the task becomes designing a suitable energy function [1] to measure the distance between the current state x_t and the editing target y. The first generation control design in energy guidance is to control the category of the generation result. It trains a classifier for $x_t$ to measure the distance between the current state $x_t$ and the target category y. However, this type of method requires additional training, and due to the influence of noise, the classification of $x_t$ is challenging. In this paper, we propose taking advantage of the strong semantics in the intermediate features of pre-trained stable diffusion, which transforms the editing target y into the feature matching between the editing starting point and the target point. Our approach eliminates training cost while achieving good editing results.
>
> [1] Diffusion models beat GANs on image synthesis
>
> - memory bank is a container to store information for visual cross-attention
>
> Although feature matching can achieve editing operations, the final result can not maintain consistency with the original image, such as in unedited areas. To maintain consistency, we use the memory bank to collect intermediate information during the image inversion and inject it into the editing process through visual cross-attention. This allows the editing process to fully capture the original image information. In Fig.9 of the original paper, we demonstrate the important role of visual cross-attention in maintaining content consistency.
>
> Therefore, these components are indispensable for our editing framework.
>
> # W2. FID compared with DragGAN
>
> This is determined by the base model. DragGAN is designed based on the pre-trained StyleGAN, while our method is based on the pre-trained Stable Diffusion. The generative capability of stable diffusion is significantly better than that of StyleGAN. The results in Fig.8 in the original paper also show that the generative capability of GAN models is limited. For unaligned images, GAN models can produce severe degradation. Even in aligned images, GANs cannot generate a good background, as shown in DragGAN’s results in our paper. In the new version of the paper, we added more discussion on DragGAN's robustness in Appendix.7.
>
> # W3. Identity by visual cross-attention
>
> We need to clarify that our method does not require training. Compared to the latest diffusion-based image editing method, DragDiffusion, our method also has performance advantages without the need for training LORA for each image like DragDiffusion. We will continue to improve content consistency in future work.
>
> # W4.&Q1. Ablation study of hyper-parameters and guidance time
>
> Thank you for this useful suggestion. We add an ablation study on the weights ($w_e$, $w_c$) of the image editing ($E_{edit}$) and content consistency ($E_{content}$) energy functions in Appendix 4. Results show that $E_{edit}$ can complete the editing operation with a small $w_e$, and the influence of $w_c$ on $E_{edit}$ is relatively small. Therefore, in our design, we apply a larger weight to $E_{content}$ to better maintain content consistency while achieving the editing goals.
>
> The ablation study on when to introduce gradient guidance shows that it is important to introduce guidance in the early sampling stages. As the sampling iterations progress, the effect of the guidance information becomes weaker. Moreover, introducing guidance information in more time steps helps improve the texture details. Finally, we introduce guidance information in the first 30 steps to balance performance and complexity.

---

> ### Author Response · Authors · 2023-11-16
>
> Dear reviewer, thanks a lot for your previous constructive comments. We would like to know if our revisions have addressed your concerns? We welcome any discussions and suggestions that will help us further improve this paper.

---

> > ### Comment · Reviewer_dwdi · 2023-11-17
> >
> > Thank you for submitting the revised version of your manuscript, along with a detailed response. I updated my score after reading these updates addressing most of the concerns. However, I have still remaining question.
> >
> > > For example, it is unclear why the FID score is higher compared to DragGAN. There is no related ablation study for that part.
> >
> > I mis-write the DragGAN on that sentance. My question was why DragonDiffusion exhibits superior FID scores compared to DragDiffusion. Given that DragDiffusion incorporates LoRA parameters, one would anticipate an enhanced performance, especially in the context of real image editing, assuming the training parameters were optimally configured.  While the incorporation of a memory bank and an energy function term is an good approach, it appears to generate the results heavily dependent on the base diffusion model. Could you offer a more detailed explanation or theoretical justification for that?

---

> ### Author Response · Authors · 2023-11-17
>
> Thank you for approving our revision.
>
> To address this issue, we explore the editing guidance of our method and DragDiffusion. Specifically, DragDiffusion selects a latent $z_t$ in the diffusion process as a learnable parameter and optimizes $z_t$ through multiple iterations. However, this optimization method only considers the editing target and ignores the impact on the diffusion process, i.e., it cannot guarantee that the optimized $z_t$ conforms to the current time step t. In contrast, our method is built on score-based gradient guidance to solve $p(z_{t-1}|z_{t}, y)$, where $y$ is the editing target. It has been proved that $p(z_{t-1}|z_{t}, y)\sim N(\mu+\eta g, \Sigma)$, where $\eta g$ is the gradient guidance produce from an energy function $p(y|z_t)$. Therefore, our guidance method is compatible with the diffusion process. The energy function (i.e., feature correspondence) in our method is also proven to be accurate and robust in measuring the feature correspondence. We believe that the difference in guidance methods is one of the reasons for the performance difference.
>
> We update our paper and demonstrate a comparison of the two editing guidance methods in Appendix 13. We show the editing effects of these two guidance methods without their content consistency design (LORA and visual cross-attention). Retaining only DDIM inversion as the generation prior of the source image. The results demonstrate that the latent optimization in DragDiffusion has a deviation impact on diffusion sampling.

---

> > ### Comment · Reviewer_dwdi · 2023-11-22
> >
> > Thank you for detailed explanation and supporting experiments. I will keep the current updated score.

---

> > > ### Author Response · Authors · 2023-11-22
> > >
> > > Thank you for your efforts and time in reviewing.

---

### Official Review · Reviewer_Fbvf · 2023-10-30

**Soundness:** 3 good
**Presentation:** 3 good
**Contribution:** 2 fair
**Rating:** 6
**Confidence:** 4

**Summary:**

This paper introduces DragonDiffusion, an image editing method that allows for Drag-style manipulation on Diffusion models. By utilizing feature correspondence, this approach transforms image editing into gradient guidance. It incorporates multi-scale guidance that takes into account both semantic and geometric alignment, as well as visual cross-attention for consistency. The proposed method demonstrates promising performance across a range of image editing tasks.

**Strengths:**

- The energy motivation that originates from classifier guidance is interesting. It motivates the design of the energy function for correspondence in diffusion models.
- The visualization figure vividly demonstrates the editing effect.

**Weaknesses:**

- The clarity of how the memory bank is meaningful is not evident in this draft. As the memory bank is proposed as a contribution, the authors should provide a more comprehensive ablation study, including both quantitative and qualitative analysis.
- How the energy design makes it works is not clear, the authors should provide more details numerical studies.
- The inference time is too slow, approximately 15.93 in Table 1, which makes the solution incomparable with dragGAN.

**Questions:**

as weakness

---

> ### Author Response · Authors · 2023-11-14
>
> Thank you for your careful review and constructive suggestions. We have modified our paper to rectify your concerns. The changes relevant to your concerns are marked in green (color will be removed in the final version).
>
> # W1. The clarity of how the memory bank is meaningful
>
> We need to clarify that the memory bank is a container we designed to store intermediate information ($z_t$, $K_t$, $V_t$) during the DDIM inversion process. These intermediate outputs will be reused to guide the subsequent editing process. In Fig.9 of the original paper, we demonstrated the importance of $K_t$ and $V_t$ (i.e., visual cross-attention) in the memory bank. We also showed the importance of generating content consistency gradients $E_{content}$ through the stored $z_t$. This intermediate information can also be generated in other ways, such as a separate generation branch. But this would result in additional computation cost. In Appendix.2 of the original paper, we conducted an ablation study on the efficiency of the memory bank. However, the connection between these two ablation parts is not tight enough. We added a brief description of Appendix.2 in the ablation study section to highlight the contribution of the memory bank.
>
> # W2. How energy design works
>
> Energy guidance is the second term in Eq.2 in our original paper, which combines the generation target y with the diffusion process as q(y|$x_t$). It has been rigorously proven in [1] for generation control in the diffusion models. To make it work, we need to design an energy function to measure the distance between the current state $x_t$ and the editing target y. The first generation control design in energy guidance is to control the category of the generation result. It trains a classifier for $x_t$ to measure the distance between the current state $x_t$ and the target category y. However, this type of method requires additional training, and due to the influence of noise, the classification on $x_t$ is challenging. In this paper, we propose taking advantage of the strong semantics of the intermediate features in pre-trained stable diffusion to transform the editing target y into the feature similarity between the editing starting point and the target point. Our approach eliminates training cost while achieving good editing results. In Fig. 4, we investigate how to make the energy guidance work better. In Fig. 5, we demonstrate how to combine different energy functions to achieve editing goals. In addition, we update more ablation studies in Appendix.3 and Appendix.4, showing the impact of more components on the working of energy guidance.
>
> To make the gradient guidance process clearer, we visualize the gradients generated by the content consistency ($E_{content}$) and editing ($E_{edit}$) energy functions at different time steps in Appendix.5. It presents a gradually converging process, and the gradient generated by each energy function mainly focuses on their respective target areas.
>
> [1] Diffusion models beat GANs on image synthesis
>
> # W3. Incomparable with DragGAN
>
> The higher inference complexity is determined by the base model. DragGAN is designed based on StyleGAN, while our method is based on Stable Diffusion. GAN models only require one inference to output results, while diffusion requires multiple iterations, resulting in a higher time complexity. However, the generative capability of the diffusion models is significantly better than that of the GANs. For example, in terms of robustness, DragGAN requires a specific trained model when editing different categories of images, and the input image cannot deviate from the distribution of GAN's training set. In Fig.8 of the original paper, we show that DragGAN's editing results have severe degradation without image alignment. In the new version of the paper, we added more discussion on DragGAN's robustness in Appendix 7. As can be seen, the GAN-based editing method fails in complex scenarios. In contrast, our method is based on the diffusion model with high robustness to edit general images. It is worth noting that the complexity of our method in the preparation stage is significantly lower than that of DragGAN (3.62s vs 52.40s in Tab.1).
>
> Most importantly, our DragonDiffusion is a general image editing method that can accomplish a range of image editing tasks, e.g., object moving, resizing, pasting, content dragging, and appearance replacing. In contrast, DragGAN is limited to content dragging.

---

> ### Author Response · Authors · 2023-11-16
>
> Dear reviewer, thanks a lot for your previous constructive comments. We would like to know if our revisions have addressed your concerns? We welcome any discussions and suggestions that will help us further improve this paper.

---

> ### Author Response · Authors · 2023-11-21
> **Looking forward to further feedbacks**
>
> Dear Reviewer Fbvf,
>
> Thank you again for your great efforts and valuable suggestions. We have carefully addressed your concerns. We hope you are satisfied with our response. As the discussion phase is about to close, we are looking forward to hearing from you about any further feedback. We will be happy to clarify further concerns (if any).
>
> Best,
>
> Authors

---

### Official Review · Reviewer_bwVZ · 2023-10-31

**Soundness:** 4 excellent
**Presentation:** 3 good
**Contribution:** 3 good
**Rating:** 6
**Confidence:** 4

**Summary:**

The paper presents a method enabling user 'dragging' style motion control image editing, similar to DragGAN but with diffusion models. To achieve this, DDIM inversion is performed first on the input image, while the intermediate features from the UNet are saved. During the forward image editing pass, starting from the DDIM-inverted noise, at each diffusion step, three terms are calculated: 1. A cosine similarity score between the dragging patch of the DDIM inversion features and current generation features to gain local consistency, 2. A cosine similarity score between the mean features of the dragging patch of the DDIM inversion features and current generation features to gain global appearance consistency, and 3. Similarity between the unchanged features. The final score gradient is calculated by perturbing the original gradient with the gradient of a weighted sum of these similarities constraints to zt.

**Strengths:**

* The task of user-defined handles is challenging and well-motivated -- – supported by various applications shown in the paper.
* Evaluations were done with reasonable metrics and against SOTA methods, and decent improvements can be observed, especially the efficiency compared with DragDiffusion. Nice qualitative results are shown.
* The method has significantly less complexity comparing with prior works, but seems to work well.

**Weaknesses:**

* Compared with prior (and concurrent) works such as DragGAN and DragDiffusion, way too few samples are shown. The paper and supplementary do not present enough challenging and diverse qualitative samples and comparisons.
* The ablation is a bit incomplete. E.g. it will be nice to see some ablations on the usefulness of S_global.
* Some flickering still happens in the no-change areas, e.g. clouds in the sun example and background in the apple example. If this is because of the balance between different losses, some ablation could be very helpful.
* The identity preservation also seems a bit off, e.g. patterns of the apple. However, I think it is a relatively minor issue as other works also cannot completely fix this issue.

**Questions:**

None

---

> ### Author Response · Authors · 2023-11-14
>
> Thank you for your careful review and constructive suggestions. We have modified our paper to rectify your concerns. The changes relevant to your concerns are marked in blue (color will be removed in the final version).
>
> # W1. More challenging and diverse qualitative samples
>
> Thanks for this useful suggestion. We add more content dragging results in complex scenarios in Appendix.7. These results further illustrate that
> - The generalization ability of DragGAN is limited by the GANs. Each trained model can only edit a specific category of images, such as face, elephant, and car. Moreover, the input image cannot deviate (e.g., unaligned faces, multiple faces or cars) from the training distribution of the corresponding GAN model. Otherwise, obvious degradation will occur.
> - Our method performs better than DragDiffusion without the need for training. To analyze the performance differences, we further discuss the differences in guidance methods between our approach (score-based guidance) and DragDiffusion (latent optimization) in Appendix.13. The results show that treating $z_t$ as a learnable variable in DragDiffusion can affect the original iteration distribution of the diffusion model.
>
> The most important is that both DragGAN and DragDiffusion are designed specifically for content dragging. Our method is a general editing framework, not limited to content dragging. Due to the previous version of the paper focusing too much on content dragging in the experiment, narrowing down this contribution (as pointed out by R1). We have reformulated the experiment section, demonstrating the editing capabilities for various tasks, further highlighting that our approach is a general and fine-grained image editing framework.
>
> # W2. Ablation of $S_{global}$
>
> In our method, $S_{global}$ controls the appearance of a certain area to be consistent with a reference area. It is used in the inpainting of the object moving task and the appearance control in the appearance replacing task. Indeed, the previous version lacked an experiment to show the effectiveness of $S_{global}$. We add an ablation study for $S_{global}$ in Appendix.3, demonstrating its appearance control in object moving and appearance replacing.
>
> # W3. Ablation of weights for different energy functions
>
> Yes, there is a trade-off between different energy functions. The energy functions in our method mainly include $E_{edit}$ and $E_{content}$, designed for editing and content consistency, respectively. We add an ablation study for the weights ($w_e$, $w_c$) of these two functions in Appendix.4. Results show that $E_{edit}$ can complete the editing operation with a small w_e, and the influence of $w_c$ on $E_{edit}$ is relatively small. Therefore, in our design, we apply a larger weight to $E_{content}$ to better maintain content consistency while achieving the editing goals. In our demo video, we directly connect the results of different editing operations into a video, which may amplify some subtle jitter.
>
> # W4. Identity preservation
>
> Yes, some details may have some deviations. Our method does not require training, and maintaining high consistency is indeed a challenge. We will continue to improve in future work.

---

> ### Author Response · Authors · 2023-11-16
>
> Dear reviewer, thanks a lot for your previous constructive comments. We would like to know if our revisions have addressed your concerns? We welcome any discussions and suggestions that will help us further improve this paper.

---

> > ### Comment · Reviewer_bwVZ · 2023-11-22
> >
> > Thanks to the authors for their replies. My concerns are addressed and I will maintain my current score.

---

> > > ### Author Response · Authors · 2023-11-22
> > >
> > > Thank you for your efforts and time in reviewing.

---

### Official Review · Reviewer_vgvS · 2023-11-01

**Soundness:** 4 excellent
**Presentation:** 3 good
**Contribution:** 3 good
**Rating:** 6
**Confidence:** 5

**Summary:**

This paper tackles a suite of image editing task (dragging editing in particular) via gradient guidance to drive the sampling from the inversed latent towards the editing target. To ensure consistency between the editing output and the original input, the KV values are cached to form a memory bank that retains the semantics of the original image. The gradient guidance formulation can deal with multiple tasks like dragging edit, object removal, object resizing, appearance replacing and object pasting. The method is mainly compared with prior dragging-based image editing approaches and shows improved quality. The results on other image tasks are also quite impressive.

**Strengths:**

- While gradient guidance has been explored extensively, using this idea as a general approach to accomplish multiple image editing tasks is cool. The qualitative results as shown in Figure 12 is stunning. And, all of these are achieved without the use of any auxiliary model.

- Caching the memory bank for improved image information preservation is a useful technique.

- It is welcome to report the detailed model inference time as shown in Table 1.

- The proposed method works well for real images, while the editing on real images is usually challenging for many prior methods.

**Weaknesses:**

- First of all, the paper proposes to use gradient guidance sampling for a bunch of tasks, but the paper writing and the experiments mainly focus on dragging-based editing. This will narrow down the scope of the paper quite a lot. It is suggested to formulate the paper as a general solution and equally treat multiple tasks.

- Also, the experiment is not thorough enough. It is suggested to conduct comparisons on other tasks besides dragging edit. For example, for object pasting, it is suggested to compare against the work "paint by example". For appearance replacement, it is suggested to compare against "Diffusion Self-Guidance for Controllable Image Generation" and "Null-text Inversion".

- There are some typos in the paper. For example, Equation 2 is not correct.

- There is no limitation analysis for the proposed method.

**Questions:**

I'm glad to see more comprehensive comparison against more approaches as aforementioned.

---

> ### Author Response · Authors · 2023-11-14
>
> Thank you for your careful review and constructive suggestions. To rectify your concerns, we have modulated our paper. The changes relevant to your concerns are marked in red (color will be removed in the final version).
>
> # W1.&W2. More comparisons
>
> Thank you for this suggestion to help us highlight the contributions of this paper. A minor clarification is that Self-Guidance was not open-sourced before the ICLR submission, so the initial version did not include a comparison with it. In our new version, we divide Section 4.1 into two parts: content dragging comparison and comparisons on other tasks. In the updated part, we demonstrate our method against Self-Guidance and Paint-by-example in object moving, appearance replacing, and object pasting tasks. We add more comparisons in Appendix.8.
>
> Moreover, we further discuss the differences between our method and text-guided editing methods in Appendix.9 by comparing with Null-text inversion, InstructPix2Pix, and Self-Guidance.
>
> - Compare with Self-Guidance. Self-Guidance is a text-guided editing method. Due to the coarse-grained correspondence between text and image content, it can only achieve object-level editing and is difficult to achieve editing objectives in some complex scenes or multi-object scenarios. In addition, Self-Guidance lacks consistency constraints, which leads to deviations between the editing results and the original image. In comparison, the impact of text on our method is minimal, as shown in Appendix.4. It shows that the editing results produced by different texts (relevant or irrelevant) are close and meet the editing requirements.
> - Compare with Paint-by-example. Paint-by-example is a learnable object pasting method. We must admit that such learnable methods can produce more natural pasting effects. However, encoding the object image into multiple tokens can disrupt the object's identity, leading to changes in the identity of the editing results. In contrast, our method adds guidance at each diffusion step, resulting in better identity consistency in the editing results.
> - Compare with other text-guided methods (e.g., Null-text inversion, InstructPix2Pix). Although these methods can also achieve object modulation, the editing results have no reference and randomly dependent on text description. In addition, similar to Self-Guidance, text has difficulty achieving a good correspondence with images in complex scenes and multi-object scenarios, leading to editing failures.
>
> It is worth noting that most of these compared methods can only accomplish one task, while we can complete these tasks with a single framework. Through these comparisons, we want to demonstrate that our method is a general fine-grained image editing framework with promising performance.
>
> Another related change is that we remove the original Application section and present the application visualization as the teaser on the first page to reduce redundant content.
>
> # W3. Typos
>
> Thanks for the careful review. We have corrected Equation 2.
>
> # W4. Limitations
>
> Thanks for this suggestion. We add a discussion of limitations in Appendix.12. It mainly includes that there is still room to further improve the inference speed, and there are some hyperparameters in our training-free pipeline that can affect the editing results.

---

> ### Author Response · Authors · 2023-11-16
>
> Dear reviewer, thanks a lot for your previous constructive comments. We would like to know if our revisions have addressed your concerns? We welcome any discussions and suggestions that will help us further improve this paper.

---

> > ### Comment · Reviewer_vgvS · 2023-12-04
> > **Official Comment by Reviewer vgvS**
> >
> > Thanks for the authors' response. The paper has been improved and the rebuttal solves my questions. I will maintain my original rating.

---

### Comment · Area_Chair_9Q66 · 2023-11-15
**Please engage in reviewer-author discussion**

Dear reviewers,

The paper got diverging scores. The authors have provided their response to the comments.
Could you look through the other reviews and engage into the discussion with authors? See if their response changes your assessment of the submission?

Thanks!
AC

---

### Meta-Review · Area_Chair_9Q66 · 2023-12-06

**Metareview:**

Summary
This work presents a general training-free framework for drag-style image editing tasks. They are achieved by leveraging gradient guidance to drive the sampling process and visual cross-attention based on memory bank to ensure consistency. Impressive performance on various editing tasks are shown in the experiments.

Strengths
All reviewers think the editing performance of the proposed method is impressive. Compared to existing works such as DragGAN, DragDiffusion and Self-Guidance, the editing capability of this work is more powerful in working across several tasks withouth training.
The effectiveness of the proposed components is also validated through ablation study.

Weaknesses
There is still a problem with this method in identity preservation and flickering effects. In addition, the writing should be further improved.

**Justification For Why Not Higher Score:**

This method still suffers from the issues of the identity preservation and flickering effect. In addition, as mentioned by several reviewers the writing should be further improved.

**Justification For Why Not Lower Score:**

All reviwers agree that this work presents a powerful general drag-style editing framework without training, which is interesting and impressive. The effectiveness of the proposed components is well analyzed and validated through experiments.

---

### Decision · Program_Chairs · 2024-01-16

Accept (spotlight)